# A Generalized Alternating Method for Bilevel Optimization under the Polyak-Łojasiewicz Condition

**Quan Xiao**
Rensselaer Polytechnic Institute
Troy, NY, USA
xiaoq5@rpi.edu

**Songtao Lu**
IBM Research
Yorktown Heights, NY, USA
songtao@ibm.com

**Tianyi Chen**
Rensselaer Polytechnic Institute
Troy, NY, USA
chentianyi19@gmail.com

## Abstract

Bilevel optimization has recently regained interest owing to its applications in emerging machine learning fields such as hyperparameter optimization, meta-learning, and reinforcement learning. Recent results have shown that simple alternating (implicit) gradient-based algorithms can match the convergence rate of single-level gradient descent (GD) when addressing bilevel problems with a strongly convex lower-level objective. However, it remains unclear whether this result can be generalized to bilevel problems beyond this basic setting. In this paper, we first introduce a stationary metric for the considered bilevel problems, which generalizes the existing metric, for a nonconvex lower-level objective that satisfies the Polyak-Łojasiewicz (PL) condition. We then propose a Generalized ALternating mEthod for bilevel opTimization (GALET) tailored to BLO with convex PL LL problem and establish that GALET achieves an $\epsilon$-stationary point for the considered problem within $\tilde{\mathcal{O}}(\epsilon^{-1})$ iterations, which matches the iteration complexity of GD for single-level smooth nonconvex problems.

## 1 Introduction

Bilevel optimization (BLO) is a hierarchical optimization framework that aims to minimize the upper-level (UL) objective, which depends on the optimal solutions of the lower-level (LL) problem. Since its introduction in the 1970s [5], BLO has been extensively studied in operations research, mathematics, engineering, and economics communities [14], and has found applications in image processing [12] and wireless communications [8]. Recently, BLO has regained interests as a unified framework of modern machine-learning applications, including hyperparameter optimization [49, 21, 22, 54], meta-learning [20], representation learning [3], reinforcement learning [62, 59], continual learning [55, 4], adversarial learning [73] and neural architecture search [39]; see [41].

In this paper, we consider BLO in the following form

$$\min_{x \in \mathbb{R}^{d_x}, y \in S(x)} \quad f(x, y) \qquad \text{s.t. } S(x) \triangleq \arg\min_{y \in \mathbb{R}^{d_y}} \ g(x, y) \tag{1}$$

where both the UL objective $f(x, y)$ and LL objective $g(x, y)$ are differentiable, and the LL solution set $S(x)$ is not necessarily a singleton. For ease of notation, we denote the optimal function value of the LL objective as $g^*(x) := \min_y g(x, y)$ and call it *value function*.

Although BLO is powerful for modeling hierarchical decision-making processes, solving generic BLO problems is known to be NP-hard due to their nested structure [31]. As a result, the majority of recent works in optimization theory of BLO algorithms are centered on nonconvex UL problems with strongly convex LL problems (nonconvex-strongly-convex), which permit the development of efficient algorithms; see e.g. [24, 27, 32, 9, 35, 10, 34, 67, 66]. The strong convexity assumption for LL ensures the uniqueness of the minimizer and a simple loss landscape of the LL problem, but it

37th Conference on Neural Information Processing Systems (NeurIPS 2023).

| | **GALET** | V-PBGD | BOME | MGBiO | BGS | IAPTT-GM | IGFM |
|---|---|---|---|---|---|---|---|
| $g(x,y)$ | PL + C | PL | PL | SC | Morse-Bott | Regular | PL + C |
| Non-singleton $S(x)$ | ✓ | ✓ | ✓ | ✗ | ✓ | ✓ | ✓ |
| Provable CQ | ✓ | Relaxed | ✗ | / | / | Relaxed | / |
| Complexity | $\tilde{\mathcal{O}}(\epsilon^{-1})$ | $\tilde{\mathcal{O}}(\epsilon^{-1.5})$ | $\tilde{\mathcal{O}}(\epsilon^{-4})$ | $\tilde{\mathcal{O}}(\epsilon^{-1})$ | ✗ | ✗ | $\mathrm{poly}(\epsilon^{-1})$ |

Table 1: Comparison of the proposed method GALET with the existing BLO for non-strongly-convex LL problem (V-PBGD [57], BOME [37], MGBiO [28], BGS [2], IAPTT-GM [43], IGFM [7]). The notation $\tilde{\mathcal{O}}$ omits the dependency on $\log(\epsilon^{-1})$ terms and $\mathrm{poly}(\epsilon^{-1})$ hides the dependency worse than $\mathcal{O}(\epsilon^{-4})$. 'C', 'SC' and 'Regular' stand for convex, strongly convex and Assumption 3.1 in [43], respectively. PL, Lipschitz Hessian and the assumption that eigenvalue bounded away from $0$ in MGBiO imply SC. 'Relaxed' means that they solve a relaxed problem without CQ-invalid issue and '/' means that CQ is not needed as it is based on the implicit function theorem.

excludes many intriguing applications of BLO. In the context of machine learning, the LL objective might represent the training loss of a neural network, which can be non-strongly-convex [65].

To measure the efficiency of solving nonconvex BLO problems, it is essential to define its stationarity, which involves identifying a necessary condition for the optimality of BLO. In cases where the LL problem exhibits strongly convexity, the solution set $S(x)$ becomes a singleton, leading to a natural definition of stationarity as the stationary point of the overall objective $f(x, S(x))$, i.e., $\nabla f(x, S(x)) = 0$. However, for BLO problems with non-strongly-convex LL problems, $S(x)$ may have multiple solutions, rendering $\nabla f(x, S(x))$ ill-posed. This motivates an intriguing question:

**Q1: What is a good metric of stationarity for BLO with nonconvex LL problems?**

To address this question, we focus on the setting of BLO with the LL objective that satisfies the PL condition (nonconvex-PL). The PL condition not only encompasses the strongly convexity condition [24, 27, 32, 9, 35, 10, 34, 67, 66] and the Morse-Bott condition [2], but also covers many applications such as overparameterized neural networks [38], learning LQR models [19], and phase retrieval [61].

By reformulating the LL problem by its equivalent conditions, one can convert the BLO problem to a constrained optimization problem. Then with certain constraint qualification (CQ) conditions, a natural definition of the stationarity of the constrained optimization problem is the Karush–Kuhn–Tucker (KKT) point [11]. For example, constant rank CQ (CRCQ) was assumed in [37], and linear independence CQ (LICQ) was assumed or implied in [44, 28]. However, it is possible that none of these conditions hold for nonconvex-PL BLO (Section 2.2). In Section 2, we study different CQs on two constrained reformulations of (1) and then identify the best combination. Based on the right CQ on the right constrained reformulation, we prove the inherent CQ and propose a new notion of stationarity for the nonconvex-PL BLO, which strictly extends the existing measures in nonconvex-strongly-convex BLO [24, 27, 32, 9] and nonconvex-nonconvex BLO [2, 37, 28] without relaxing the problem [36, 57]. We emphasize the importance of defining new metric in Section 2.3.

Given a stationary metric, while $\epsilon$-stationary point can be found efficiently in $\mathcal{O}(\epsilon^{-1})$ iterations for nonconvex and smooth single-level problem [6], existing works on the BLO with non-strongly-convex LL problem either lack complexity guarantee [2, 43, 36, 42], or occur slower rate [57, 37, 58, 7]. Moreover, most existing algorithms update the UL variable $x$ after obtaining the LL parameter $y$ sufficiently close to the optimal set $S(x)$ by running GD from scratch, which is computationally expensive [37, 57]. In contrast, the most efficient algorithm for nonconvex-strongly-convex BLO updates $x$ and $y$ in an alternating manner, meaning that $x$ is updated after a constant number of $y$ updates from their previous values [9, 32]. Then another question arises:

**Q2: Can alternating methods achieve the $\mathcal{O}(\epsilon^{-1})$ complexity for non-strongly-convex BLO?**

Addressing this question is far from trivial. First, we need to characterize the drifting error of $S(x)$ induced by the alternating strategy, which involves the change in the LL solution sets between two consecutive UL iterations. However, we need to generalize the analysis in nonconvex-strongly-convex BLO [9, 32, 24, 27] because $S(x)$ is not a singleton. Moreover, we need to select an appropriate Lyapunov function to characterize the UL descent, as the nature candidate $f(x, S(x))$ is ill-posed without a unique LL minimizer. Finally, since the Lyapunov function we choose for UL contains both $x$ and $y$, it is crucial to account for the drifting error of $y$ as well.

By exploiting the smoothness of the value function $g^*(x)$ and with the proper design of the Lyapunov function, we demonstrate the $\tilde{\mathcal{O}}(\epsilon^{-1})$ iteration complexity of our algorithm, which is optimal in terms

of $\epsilon$. This result not only generalizes the convergence analysis in nonconvex-strongly-convex BLO [24, 27, 32, 9, 35, 10, 34, 67, 66] to the broader problem class, but also improves the complexity of existing works on *nonconvex-non-strongly-convex* BLO, specifically $\tilde{\mathcal{O}}(\epsilon^{-1.5})$ in [57] and $\tilde{\mathcal{O}}(\epsilon^{-4})$ in [37]; see Table 1. We present our algorithm in Section 3 and analyze its convergence in Section 4, followed by the simulations and conclusions in Section 5.

## 1.1 Related works

**Nonconvex-strongly-convex BLO.** The interest in developing efficient gradient-based methods and their nonasymptotic analysis for nonconvex-strongly-convex BLO has been invigorated by recent works [24, 32, 27, 9]. Based on the different UL gradient approximation techniques they use, these algorithms can be categorized into iterative differentiation and approximate implicit differentiation-based approaches. The iterative differentiation-based methods relax the LL problem by a dynamical system and use the automatic differentiation to approximate the UL gradient [21, 22, 25], while the approximate implicit differentiation-based methods utilize the implicit function theory and approximate the UL gradient by the Hessian inverse (e.g. Neumann series [9, 24, 27]; kernel-based methods [26]) or Hessian-vector production approximation methods (e.g. conjugate gradient descent [32, 54], gradient descent [35, 1]). Recent advances include variance reduction and momentum based methods [34, 67, 13]; warm-started BLO algorithms [1, 35]; distributed BLO approaches [63, 46, 68]; and algorithms solving BLO with constraints [64, 66]. Nevertheless, none of these attempts tackle the BLO beyond the strongly convex LL problem.

**Nonconvex-nonconvex BLO.** While nonconvex-strongly-convex BLO has been extensively studied in the literature, efficient algorithms for BLO with nonconvex LL problem remain under-explored. Among them, Liu et al. [43] developed a BLO method with initialization auxiliary and truncation of pessimistic trajectory; and Arbel and Mairal [2] generalized the implicit function theorem to a class of nonconvex LL functions and introduced a heuristic algorithm. However, these works primarily focus on analyzing the asymptotic performance of their algorithms, without providing finite-time convergence guarantees. Recently, Liu et al. [37] proposed a first-order method and established the first nonasymptotic analysis for non-strongly-convex BLO. Nonetheless, the assumptions such as CRCQ and bounded $|f|, |g|$ are relatively restrictive. Huang [28] has proposed a momentum-based BLO algorithm, but the assumptions imply strongly convexity. Another research direction has addressed the nonconvex BLO problem by relaxation, such as adding regularization in the LL [42, 50], or replacing the LL optimal solution set with its $\epsilon$-optimal solutions [36, 57]. Although this relaxation strategy overcomes the CQ-invalid issues, it introduces errors in the original bilevel problem [7]. To the best of our knowledge, none of these BLO algorithms handling multiple LL solutions can achieve the optimal iteration complexity in terms of $\epsilon$.

**Nonconvex-convex BLO.** Another line of research focuses on the BLO with convex LL problem, which can be traced back to [48, 18]. Convex LL problems pose additional challenges of multiple LL solutions which hinder from using implicit-based approaches for nonconvex-strongly-convex BLO. To tackle multiple LL solutions, an aggregation approach was proposed in [56, 40]; a primal-dual algorithm was considered in [58]; a difference-of-convex constrained reformulated problem was explored in [72, 23]; an averaged multiplier method was proposed in [45]. Recently, Chen et al. [7] has pointed out that the objective of the non-strongly-convex LL problem can be discontinuous and proposed a zeroth-order smoothing-based method; Lu and Mei [47] have solved it by penalized min-max optimization. However, none of these attempts achieve the iteration complexity of $\tilde{\mathcal{O}}(\epsilon^{-1})$. Moreover, although some works adopted KKT related concept as stationary measure, they did not find the inherent CQ condition, so the necessity of their measure to the optimality of BLO is unknown [45, 47]. In this sense, our work is complementary to them. The comparison with closely related works is summarized in Table 1.

**Notations.** For any given matrix $A \in \mathbb{R}^{d \times d}$, we list the singular values of $A$ in the increasing order as $0 \leq \sigma_1(A) \leq \sigma_2(A) \leq \cdots \leq \sigma_d(A)$ and denote the smallest positive singular value of $A$ as $\sigma_{\min}^+(A)$. We also denote $A^{-1}, A^\dagger, A^{1/2}$ and $A^{-1/2}$ as the inverse of $A$, the Moore-Penrose inverse of $A$ [29], the square root of $A$ and the square root of the inverse of $A$, respectively. $\mathrm{Ker}(A) = \{x : Ax = 0\}, \mathrm{Ran}(A) = \{Ax\}$ denotes the null space and range space of $A$.

## 2 Stationarity Metric of Nonconvex-PL BLO

We will first introduce the equivalent constraint-reformulation of the nonconvex-PL BLO and then introduce our stationarity metric, followed by a section highlighting the importance of our results.

## 2.1 Equivalent constraint-reformulation of BLO

By viewing the LL problem as a constraint to the UL problem, BLO can be reformulated as a single-level nonlinear constrained optimization problem. Based on different equivalent characteristics of the LL problem, two major reformulations are commonly used in the literature [16]. The first approach is called value function-based reformulation, that is

$$\min_{x \in \mathbb{R}^{d_x}, y \in \mathbb{R}^{d_y}} f(x,y) \quad \text{s.t.} \ \ g(x,y) - g^*(x) = 0. \tag{2}$$

Clearly, $g(x,y) - g^*(x) = 0$ is equivalent to $y \in S(x)$ so that (2) is equivalent to (1).

On the other hand, recall the definition of PL function below, which is not necessarily strongly convex or even convex [33].

**Definition 1** (**PL condition**). *The function $g(x, \cdot)$ satisfies the PL condition if there exists $\mu_g > 0$ such that for any given $x$, it holds that $\|\nabla_y g(x,y)\|^2 \geq 2\mu_g(g(x,y) - g^*(x)), \ \forall y$.*

According to Definition 1, for PL functions, $\nabla_y g(x,y) = 0$ implies $g(x,y) = g^*(x)$. Therefore, the second approach replaces the LL problem with its stationary condition, that is

$$\min_{x \in \mathbb{R}^{d_x}, y \in \mathbb{R}^{d_y}} f(x,y) \quad \text{s.t.} \ \ \nabla_y g(x,y) = 0. \tag{3}$$

We call (3) the gradient-based reformulation. The formal equivalence of (2) and (3) with (1) is established in Theorem 3 in Appendix.

For constrained optimization, the commonly used metric of quantifying the stationarity of the solutions are the KKT conditions. However, the local (resp. global) minimizers do not necessarily satisfy the KKT conditions [11]. To ensure the KKT conditions hold at local (resp. global) minimizer, one needs to assume CQ conditions, e.g., the Slater condition, LICQ, Mangasarian-Fromovitz constraint qualification (MFCQ), and the CRCQ [30]. Nevertheless, Ye and Zhu [70] have shown that none of these standard CQs are valid to the reformulation (2) for all types of BLO.

## 2.2 Stationarity metric

We will next establish the necessary condition for nonconvex-PL BLO via the calmness condition [11, Definition 6.4.1], which is weaker than the Slater condition, LICQ, MFCQ and CRCQ.

**Definition 2** (**Calmness**). *Let $(x^*, y^*)$ be the global minimizer of the constrained problem*

$$\min_{x,y} f(x,y) \quad \text{s.t.} \ \ h(x,y) = 0. \tag{4}$$

*where $h: \mathbb{R}^{d_x + d_y} \to \mathbb{R}^d$ and $d \geq 1$. If there exist positive $\epsilon$ and $M$ such that for any $q \in \mathbb{R}^d$ with $\|q\| \leq \epsilon$ and any $\|(x', y') - (x^*, y^*)\| \leq \epsilon$ which satisfies $h(x', y') + q = 0$, one has*

$$f(x', y') - f(x^*, y^*) + M\|q\| \geq 0 \tag{5}$$

*then the problem* (4) *is said to be calm with $M$.*

The calmness of a problem quantifies the sensitivity of the objective to the constraints. Specifically, the calmness conditions of reformulations (2) and (3) are defined by setting $h(x,y) = g(x,y) - g^*(x)$ and $h(x,y) = \nabla_y g(x,y)$, respectively.

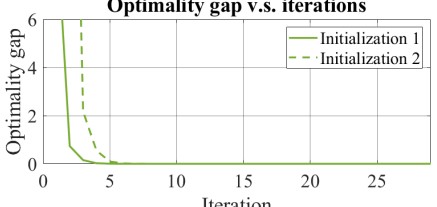

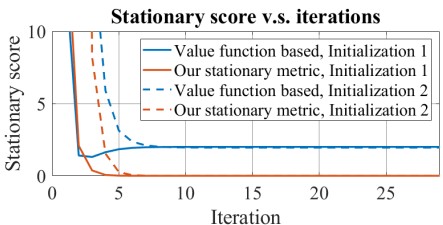

Figure 1: Example 1 under different initialization. The solid and dashed lines represent two initialization in both plots. **Top**: the distance to the global optimal set measured by (46) v.s. iteration. **Bottom**: the stationary score of (2) and (3) v.s. iteration.

The calmness condition is the weakest CQ which can be implied by Slater condition, LICQ, MFCQ and CRCQ [11]. However, as we will show, even the calmness condition does not hold for the nonconvex-PL BLO when employing the value function-based reformulation (2).

**Example 1.** *Considering $x \in \mathbb{R}, y = [y_1, y_2]^\top \in \mathbb{R}^2$, and the BLO problem as*

$$\min_{x,y} f(x,y) = x^2 + y_1 - \sin(y_2) \quad \text{s.t.} \ \ y \in \arg\min_y g(x,y) = \frac{1}{2}(x + y_1 - \sin(y_2))^2. \tag{6}$$

**Lemma 1.** *Considering the BLO problem in Example 1, the LL objective satisfies the PL condition and the global minimizers of it are within the set*

$$\{(\bar{x}, \bar{y}) \mid \bar{x} = 0.5, \bar{y} = [\bar{y}_1, \bar{y}_2]^\top, 0.5 + \bar{y}_1 - \sin(\bar{y}_2) = 0\} \tag{7}$$

*but the calmness condition of reformulation (2) is not satisfied under on all of the global minimizers.*

Figure 1 illustrates this fact by showing that the KKT score on the value function-based reformulation does not approach $0$ as the distance to the optimal set decreases. This observation implies that the KKT conditions associated with the value function-based reformulation (2) do not constitute a set of necessary conditions for global minimizers for the nonconvex-PL BLO problems. As a result, the KKT point of (2) is unable to serve as the stationary point for the nonconvex-PL BLO problems.

Therefore, we adopt the gradient-based reformulation (3). Unfortunately, it is still possible that the standard CQs do not hold for some of the nonconvex-PL BLO problems associated with the gradient-based reformulation (3). To see this, let $(x^*, y^*)$ be the global minimizer of (3). By denoting the matrix concatenating the Hessian and Jacobian as $[\nabla^2_{yy}g(x^*, y^*), \nabla^2_{yx}g(x^*, y^*)]$, the generic CQ conditions are instantiated in the gradient-based reformulation (3) by

- LICQ, MFCQ: The rows of $[\nabla^2_{yy}g(x^*, y^*), \nabla^2_{yx}g(x^*, y^*)]$ are linearly independent; and,
- CRCQ: $\exists$ neighborhood of $(x^*, y^*)$ such that $[\nabla^2_{yy}g(x, y), \nabla^2_{yx}g(x, y)]$ is of constant rank.

If $g(x, y)$ is strongly convex over $y$, then $\nabla^2_{yy}g(x, y)$ is of full rank for any $x$ and $y$, which ensures the LICQ, MFCQ and CRCQ of the gradient-based reformulation (3). However, if $g(x, y)$ merely satisfies the PL condition like **Example** 1, none of the standard CQs hold, which is established next.

**Lemma 2.** *Example 1 violates Slater condition, LICQ, MFCQ and CRCQ conditions of* (3).

Remarkably, the following lemma demonstrates that the gradient-based reformulation of the nonconvex-PL BLO inherits the calmness condition.

**Lemma 3** (**Calmness of nonconvex-PL BLO**). *If $g(x, \cdot)$ satisfies the PL condition and is smooth, and $f(x, \cdot)$ is Lipschitz continuous, then (3) is calm at its global minimizer $(x^*, y^*)$.*

The Lipschitz smoothness of $g(x, y)$ and Lipschitz continuity of $f(x, y)$ over $y$ are standard in BLO [9, 27, 24, 32, 35, 34, 1, 13, 37]. In this sense, nonconvex-PL BLO associated with the gradient-based reformulation (3) is naturally calm so that the KKT conditions are necessary conditions for its optimality [11]. A summary and illustration of our theory is shown in Figure 2.

To benefit the algorithm design (Section 3.2), we establish the necessary conditions of the optimality for nonconvex-PL BLO by slightly modifying the KKT conditions of (3) in the next theorem.

---

**Theorem 1** (**Necessary condition in nonconvex-PL BLO**). *If $g(x, \cdot)$ satisfies the PL condition and is smooth, and $f(x, \cdot)$ is Lipschitz continuous, then $\exists w^* \neq 0$ such that*

$$\mathcal{R}_x(x^*, y^*, w^*) := \|\nabla_x f(x^*, y^*) + \nabla^2_{xy}g(x^*, y^*)w^*\|^2 = 0 \tag{8a}$$

$$\mathcal{R}_w(x^*, y^*, w^*) := \|\nabla^2_{yy}g(x^*, y^*)\left(\nabla_y f(x^*, y^*) + \nabla^2_{yy}g(x^*, y^*)w^*\right)\|^2 = 0 \tag{8b}$$

$$\mathcal{R}_y(x^*, y^*) := g(x^*, y^*) - g^*(x^*) = 0 \tag{8c}$$

*hold at the global minimizer $(x^*, y^*)$ of (1).*

---

This necessary condition is tight in the sense that it is a generalization of stationary measures in the existing literature for BLO with LL problem exhibiting strongly convexity [9, 27, 24, 32], PL with CRCQ [37], invertible Hessian and singleton solution [28] and Morse-Bott functions [2]. Thanks to the inherent calmness of PL BLO, our result eliminates the CRCQ condition in [37]. We next show the connections of our results with other works.

**Nonconvex-strongly-convex BLO or PL with invertible Hessian and singleton solution.** As $S(x)$ is singleton and $\nabla_{yy}g(x, y)$ is always non-singular, the solution to (8b) is uniquely given by

$$w^* = -\left(\nabla^2_{yy}g(x^*, S(x^*))\right)^{-1}\nabla_y f(x^*, S(x^*)). \tag{9}$$

Therefore, the necessary condition in (8) is equivalent to

$$\nabla f(x^*, S(x^*)) = \nabla_x f(x^*, S(x^*)) - \nabla^2_{xy}g(x^*, S(x^*))\left(\nabla^2_{yy}g(x^*, S(x^*))\right)^{-1}\nabla_y f(x^*, S(x^*)) = 0$$

where the first equality is obtained by the implicit function theorem. Therefore, (8) recovers the necessary condition $\nabla f(x^*, S(x^*)) = 0$ for nonconvex-strongly-convex BLO.

**Nonconvex-Morse-Bott BLO.** Morse-Bott functions are special cases of PL functions [2]. In this case, $\nabla_{yy}g(x, y)$ can be singular so that (8b) may have infinite many solutions, which are given by

$$\mathcal{W}^* = - \left(\nabla_{yy}^2 g(x^*, y^*)\right)^\dagger \nabla_y f(x^*, y^*) + \text{Ker}(\nabla_{yy}^2 g(x^*, y^*)).$$

According to [2, Proposition 6], for any $y^* \in S(x^*)$, $\nabla_{xy}^2 g(x^*, y^*) \subset \text{Ran}(\nabla_{yy}^2 g(x^*, y^*))$, which is orthogonal to $\text{Ker}(\nabla_{yy}^2 g(x^*, y^*))$. As a result, although the solution to (8b) is not unique, all of possible solutions yield the unique left hand side value of (8a). i.e. $\forall w^* \in \mathcal{W}^*$,

$$\nabla_{xy}^2 g(x^*, y^*) w^* = -\nabla_{xy}^2 g(x^*, y^*) \left(\nabla_{yy}^2 g(x^*, y^*)\right)^\dagger \nabla_y f(x^*, y^*).$$

Plugging into (8a), we arrive at

$$\nabla_x f(x^*, y^*) \underbrace{-\nabla_{xy}^2 g(x^*, y^*) \left(\nabla_{yy}^2 g(x^*, y^*)\right)^\dagger \nabla_y f(x^*, y^*)}_{:=\phi(x^*, y^*)} = 0 \quad \text{and} \quad \nabla_y g(x^*, y^*) = 0$$

where $\phi(x^*, y^*)$ is the same as the degenerated implicit differentiation in [2].

Based on (8), we can define the $\epsilon$- stationary point of the original BLO problem (1) as follows.

**Definition 3 ($\epsilon$- stationary point).** *A point $(\bar{x}, \bar{y})$ is called $\epsilon$-stationary point of (1) if $\exists \bar{w}$ such that $\mathcal{R}_x(\bar{x}, \bar{y}, \bar{w}) \le \epsilon, \mathcal{R}_w(\bar{x}, \bar{y}, \bar{w}) \le \epsilon$ and $\mathcal{R}_y(\bar{x}, \bar{y}) \le \epsilon$.*

## 2.3 The importance of necessary conditions for BLO without additional CQs

Next, we emphasize the importance of deriving the necessary condition for the optimality of the nonconvex-PL BLO without additional CQs.

On the one hand, the necessary condition for the optimality of BLO is fundamental to the algorithm design and has been investigated for a long time in the optimization community [70, 71, 15, 17], but has not yet been fully understood [69]. One of the main challenges is that traditional CQs for constrained optimization are hard to check or do not hold in BLO [16, 70]. As a result, the development of mild CQs and the identification of BLO classes that inherently satisfy these CQs are considered significant contributions to the field [70, 71, 15, 17, 69]. Among those, the calmness condition is the weakest CQ [11].

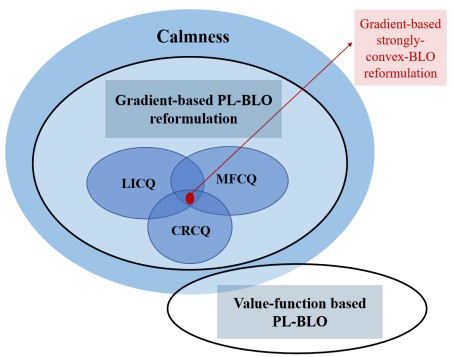

Figure 2: Illustration of our theory: relations of different CQs and BLO reformulation (2) and (3). Slater condition fails for both BLO reformulations so we do not include it.

On the other hand, recent BLO applications in machine learning often involve nonconvex LL problems, the necessary condition for the solution of which is far less explored in either optimization or machine learning community. In the optimization community, most works focus on proving linear bilevel and sequential min-max optimization satisfy certain CQs [70, 16]. In the machine learning community, works on nonconvex-nonconvex BLO either relax the LL problem that introduces error [43, 57] or impose assumptions that are hard to verify in practice such as CRCQ [37] or Morse-Bott condition [2]. To the best of our knowledge, we are the first to propose the necessary condition for the optimality of a class of nonconvex-nonconvex BLO problems with checkable assumptions. Moreover, algorithms in constrained optimization are always CQ-dependent, e.g. the convergence of an algorithm depends on whether a particular CQ condition is satisfied. As we prove that standard CQs are invalid for nonconvex-PL BLO, several existing BLO may become theoretically less grounded [37, 28].

## 3 An Alternating Method for Bilevel Problems under the PL Condition

In this section, we introduce our Generalized ALternating mEthod for bilevel opTimization with convex LL problem, GALET for short, and then elucidate its relations with the existing algorithms.

### 3.1 Algorithm development

To attain the $\epsilon$-stationary point of (1) in the sense of Definition 3, we alternately update $x$ and $y$ to reduce computational costs. At iteration $k$, we update $y^{k+1}$ by $N$-step GD on $g(x^k, y)$ with $y^{k,0} = y^k$ and $y^{k+1} = y^{k,N}$ as

$$y^{k,n+1} = y^{k,n} - \beta \nabla g(x^k, y^{k,n}) \tag{10}$$

---

**Algorithm 1** GALET for nonconvex-PL BLO

1: Initialization $\{x^0, y^0\}$, stepsizes $\{\alpha, \beta, \rho\}$
2: **for** $k = 0$ **to** $K - 1$ **do**
3:     **for** $n = 0$ **to** $N - 1$ **do**     $\triangleright y^{k,0} = y^k$
4:         update $y^{k,n+1}$ by (10)
5:     **end for**               $\triangleright y^{k+1} = y^{k,N}$
6:     **for** $t = 0$ **to** $T - 1$ **do**     $\triangleright w^{k,0} = 0$
7:         update $w^{k,t+1}$ by (12b)
8:     **end for**               $\triangleright w^{k+1} = w^{k,T}$
9:     calculate $d_x^k$ by (13)
10:    update $x^{k+1} = x^k - \alpha d_x^k$
11: **end for**

---

While setting $N = 1$ is possible, we retain $N$ for generality. We then update $w$ via the fixed-point equation derived from (8b) and employ $\nabla_{yy}g(x,y)\left(\nabla_y f(x,y) + \nabla_{yy}g(x,y)w\right)$ as the increment for $w$. This increment can be viewed as the gradient of $\mathcal{L}(x,y,w)$ defined as

$$\mathcal{L}(x,y,w) := \frac{1}{2}\left\|\nabla_y f(x,y) + \nabla_{yy}^2 g(x,y)w\right\|^2 \tag{11}$$

which is quadratic w.r.t. $w$, given $x^k$ and $y^{k+1}$.

However, unlike the LL objective $g(x,y)$, the objective $\mathcal{L}(x,y,w)$ is Lipschitz smooth with respect to $x$ and $y$ only for bounded $w$, which makes it difficult to control the change of solution (11) under different $x$ and $y$. Hence, we update $w^{k+1}$ via $T$-step GD on with $w^{k,0} = 0$ and $w^{k+1} = w^{k,T}$ as

$$w^{k,t+1} = w^{k,t} - \rho d_w^{k,t}, \tag{12a}$$

$$d_w^{k,t} := \nabla_{yy}^2 g(x^k, y^{k+1})\left(\nabla_y f(x^k, y^{k+1}) + \nabla_{yy}^2 g(x^k, y^{k+1})w^{k,t}\right). \tag{12b}$$

After obtaining the updated $y^{k+1}$ and $w^{k+1}$, we update $x^k$ by the fixed point equation of (8a) as

$$x^{k+1} = x^k - \alpha d_x^k, \quad \text{with} \quad d_x^k := \nabla_x f(x^k, y^{k+1}) + \nabla_{xy}^2 g(x^k, y^{k+1})w^{k+1}. \tag{13}$$

We summarize our algorithm in Algorithm 1. We choose $w^{k,0} = 0$ for simplicity, but $w^{k,0} = w^0 \neq 0$ is also valid. Same convergence statement can hold since the boundedness and Lipschitz continuity of limit points $\lim_{t\to\infty} w^{k,t}$ are still guaranteed. From the algorithm perspective, [2] shares similarities with us. However, without recognizing the property of GD converging to the minimal norm solution [52], Arbel and Mairal [2] introduces an additional Hessian into the objective (11), resulting in the calculation of fourth-order Hessian operation, which is more complex than GALET.

### 3.2 Relation with algorithms in nonconvex-strongly-convex BLO

We explain the relation between GALET and methods in the nonconvex-strongly-convex BLO.

First, if $g(x,y)$ is strongly convex in $y$, minimizing $\mathcal{L}(x,y,w)$ over $w$ yields the unique solution $w^*(x,y) = -\left(\nabla_{yy}^2 g(x,y)\right)^{-1}\nabla_y f(x,y)$. This means that optimizing $\mathcal{L}(x,y,w)$ corresponds to implicit differentiation in the nonconvex-strongly-convex BLO [9, 27, 24, 32]. Therefore, we refer the optimization of $w$ as the ***shadow implicit gradient*** level.

On the other hand, if $\nabla_{yy}^2 g(x,y)$ is positive definite, the problem (11) over $w$ is equivalent to

$$\min_w \left\{\frac{1}{2}w^\top \nabla_{yy}^2 g(x,y)w + w^\top \nabla_y f(x,y)\right\} \tag{14}$$

which can be verified by their first-order necessary conditions. Thus, one can update $w$ by GD on (14) via

$$d_w^{k,t} := \nabla_y f(x^k, y^{k+1}) + \nabla_{yy}^2 g(x^k, y^{k+1})w^{k,t}. \tag{15}$$

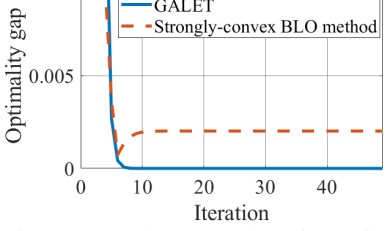

Figure 3: The $w$ update in (15) for nonconvex-strongly-convex does not work well for nonconvex-PL BLO.

Note that the increment in (15) eliminates the extra Hessian in (12b) and recovers the updates for nonconvex-strongly-convex BLO in [35, 27, 32, 9, 66]. Actually, the additional Hessian in (12b) is inevitable for an algorithm to find the stationary point of the nonconvex-PL BLO. Figure 3 shows the comparative results of our algorithm with the nonconvex-strongly-convex BLO algorithm using (15) instead of (12b) on Example 1, where the global optimality is measured by the distance to the global optimal set (7) with the explicit form stated in (46). We observe that the update (15) fails to find the global optimizer of the example in Lemma 1 so that the additional Hessian in (12b) is unavoidable.

# 4 Convergence Analysis

In this section, we provide the convergence rate analysis under convex PL LL settings. We first introduce the assumptions and the challenges. Next, we analyze the descent property and drifting error in LL, the bias of $w^{k+1}$ to the optimal solution of $\min_w \mathcal{L}(x^k, y^{k+1}, w)$ and the descent property of UL. Finally, we define a new Lyapunov function and state the iteration complexity of GALET.

**Assumption 1** (Lipschitz continuity). *Assume that $\nabla f, \nabla g$ and $\nabla^2 g$ are Lipschitz continuous with $\ell_{f,1}, \ell_{g,1}$ and $\ell_{g,2}$, respectively. Additionally, we assume $f(x, y)$ is $\ell_{f,0}$-Lipschitz continuous over $y$.*

**Assumption 2** (Landscape of LL objective). *Assume that $g(x, y)$ is $\mu_g$-PL over $y$. Moreover, let $\sigma_g > 0$ be the lower-bound of the positive singular values of Hessian, i.e. $\sigma_g = \inf_{x,y} \{\sigma^+_{\min}(\nabla^2_{yy} g(x, y))\}$.*

**Remark 1.** By definition, singular values are always nonnegative. We use $\sigma_g$ to denote the lower bound of the **non-zero** singular values of $\nabla^2_{yy} g(x, y)$. Assumption 2 means that the **non-zero** singular values are bounded away from 0 on the entire domain. Given Assumption 1 that the Hessian is globally Lipschitz, they together potentially rule out negative eigenvalues of the Hessian. However, different from strong convexity [9, 27, 24, 32, 28], this assumption still permits $\nabla^2_{yy} g(x, y)$ to possess zero eigenvalues and includes BLO problems of multiple LL solutions. As an example, this includes the loss of the (overparameterized) generalized linear model [52]. In addition, Assumption 2 is needed only in the convergence rate analysis along the optimization path to ensure the sequence $\{w^k\}$ is well-behaved. Therefore, it is possible to narrow down either the Hessian Lipschitz condition or the lower bound of singular values to a bounded region, when the local optimal sets are bounded.

*Challenges of analyzing GALET.* The absence of strong convexity of the LL brings challenges to characterize the convergence of GALET. To see this, recall that in recent analysis of BLO such as [9], when $g(x, \cdot)$ is strongly convex, the minimizer of LL is unique. Therefore, to quantify the convergence of Algorithm 1, one can use the Lyapunov function $\mathbb{V}^k_{\mathrm{SC}} := f(x^k, S(x^k)) + \|y^k - S(x^k)\|^2$ where $f(x^k, S(x^k))$ and $\|y^k - S(x^k)\|^2$ are used to account for the UL descent and LL error caused by the alternating update and the inexactness, respectively. However, $\mathbb{V}_{\mathrm{SC}}$ is not well-defined when $S(x)$ is not unique. Therefore, it is necessary to seek for a new Lyapunov function.

## 4.1 Descent of each sequence in GALET

A nature alternative of $\|y^k - S(x^k)\|^2$ under the PL condition is the LL optimality residual $\mathcal{R}_y(x^k, y^k)$, the evolution of which between two steps can be quantified by

$$\mathcal{R}_y(x^{k+1}, y^{k+1}) - \mathcal{R}_y(x^k, y^k) = \underbrace{\mathcal{R}_y(x^{k+1}, y^{k+1}) - \mathcal{R}_y(x^k, y^{k+1})}_{(17b)} + \underbrace{\mathcal{R}_y(x^k, y^{k+1}) - \mathcal{R}_y(x^k, y^k)}_{(17a)} \quad (16)$$

where the first term characterizes the drifting LL optimality gap after updating $x$, and the second term shows the descent amount by one-step GD on $y$. Unlike the strongly convex case where $S(x)$ is Lipschitz continuous, both $g^*(x)$ and $g(x, y)$ are not Lipschitz continuous, so we cannot directly bound the the first difference term in (16) by the drifting of update $\|x^{k+1} - x^k\|$. Owing to the opposite sign of $g(x, y)$ and $g^*(x)$, we bound the first term in (16) by the smoothness of $g(x, y)$ and $g^*(x)$. The second term in (16) can be easily bounded by the fact that running GD on PL function ensures the contraction of function value distance.

**Lemma 4.** *Under Assumption 1–2, let $y^{k+1}$ be the point generated by* (10) *given $x^k$ and the stepsize $\beta \leq \frac{1}{\ell_{g,1}}$. Then it holds that*

$$\mathcal{R}_y(x^k, y^{k+1}) \leq (1 - \beta\mu_g)^N \mathcal{R}_y(x^k, y^k) \quad (17a)$$

$$\mathcal{R}_y(x^{k+1}, y^{k+1}) \leq \left(1 + \frac{\alpha\eta_1\ell_{g,1}}{\mu_g}\right) \mathcal{R}_y(x^k, y^{k+1}) + \left(\frac{\alpha\ell_{g,1}}{2\eta_1} + \frac{\ell_{g,1} + L_g}{2}\alpha^2\right) \|d^k_x\|^2 \quad (17b)$$

*where $L_g := \ell_{g,1}(1 + \ell_{g,1}/2\mu_g)$ and $\eta_1 = \mathcal{O}(1)$ is a constant that will be chosen in the final theorem.*

Rearranging the terms in (17a) and using the fact that $g(x, y) \geq g^*(x)$, the second term in (16) can be upper bounded by a negative term and $\mathcal{O}(\beta^2)$. Adding (17a) and (17b), letting $\alpha, \beta$ is the same order and choosing $\eta_1 = \mathcal{O}(1)$ properly yield

$$\mathcal{R}_y(x^{k+1}, y^{k+1}) - \mathcal{R}_y(x^k, y^k) \leq -\mathcal{O}(\alpha)\mathcal{R}_y(x^k, y^k) + \mathcal{O}(\alpha)\|d^k_x\|^2. \quad (18)$$

When we choose $\eta_1$ large enough, the second term will be dominated by the negative term $-\mathcal{O}(\alpha)\|d^k_x\|^2$ given by the UL descent, so it will be canceled out.

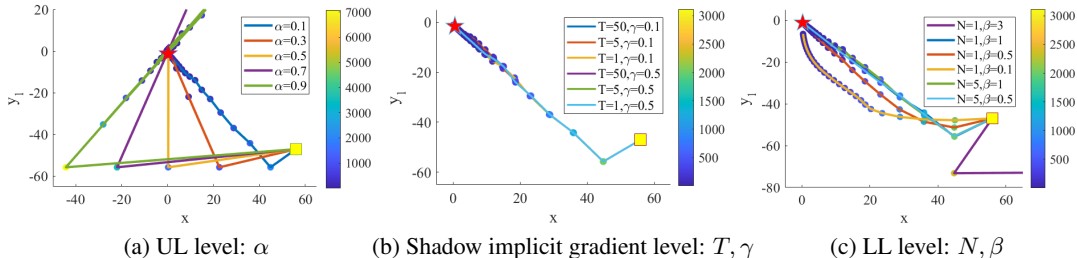

| (a) UL level: $\alpha$ | (b) Shadow implicit gradient level: $T, \gamma$ | (c) LL level: $N, \beta$ |

Figure 4: Trajectories of GALET under different choices of the parameters in UL, LM and LL level for Example 1. The $x$-axis denotes the value of $x^k$, the $y$-axis represents the first coordinates of $y^k$. The color of the point shows the optimality gap measured by (46) with the color bar shown right. The yellow square ■ is the starting point and the red star ⋆ represent the optimal $(x^*, y^*)$.

With the iterates generated by GD initialized at 0 converging to the minimal norm solution of the least squares problem [52], we can characterize the drifting error of $w$ using the Lipschitz continuity of its minimal norm solution. The following lemma characterizes the error of $w^{k+1}$ to the minimal-norm solution of the shadow implicit gradient level $w^\dagger(x, y) := -\left(\nabla_{yy}^2 g(x, y)\right)^\dagger \nabla_y f(x, y)$.

**Lemma 5.** *Under Assumption 1–2, we let $w^{k+1}$ denote the iterate generated by* (12a) *given $(x^k, y^{k+1})$ and $w^{k,0} = 0$, and denote $b_k := \|w^{k+1} - w^\dagger(x^k, y^{k+1})\|$. If we choose the stepsize $\rho \leq \frac{1}{\ell_{g,1}^2}$, then the shadow implicit gradient error can be bounded by $b_k^2 \leq 2(1 - \rho \sigma_g^2)^T \ell_{f,0}^2 / \mu_g$.*

Instead of using $f(x, S(x))$ which is ill-posed, we integrate the increment of $x$ to get $F(x, y; w) = f(x, y) + w^\top \nabla_y g(x, y)$. Then by plugging the optimal minimal norm solution $w^\dagger(x, y)$, we choose $F(x^k, y^k; w^\dagger(x^k, y^k))$ as the counterpart of $f(x^k, S(x^k))$ in $\mathbb{V}_{SC}^k$. We can quantify its difference between two adjacent steps by the smoothness of $F(x, y; w)$.

**Lemma 6.** *Under Assumption 1–2, by choosing $\alpha \leq \min\left\{\frac{1}{4L_F + 8L_w \ell_{g,1}}, \frac{\beta \eta_2}{2L_w^2}\right\}$, one has*

$$F(x^{k+1}, y^{k+1}; w^\dagger(x^{k+1}, y^{k+1})) - F(x^k, y^k; w^\dagger(x^k, y^k))$$
$$\leq -\frac{\alpha}{8}\|d_x^k\|^2 + \frac{\alpha \ell_{g,1}^2}{2}b_k^2 + \frac{\beta(\eta_2 \ell_{g,1}^2 + 2NL_w \ell_{g,1}^2 + 2\ell_{f,0}\ell_{g,2}) + L_F \ell_{g,1}^2 \beta^2}{\mu_g}\mathcal{R}_y(x^k, y^k) \quad (19)$$

*where $L_w$ is the constant defined in Appendix and constant $\eta_2$ will be chosen in the final theorem.*

### 4.2 A new Lyapunov function

Based on (18) and (19), we can define a new Lyapunov function for the nonconvex-PL BLO as

$$\mathbb{V}^k := F(x^k, y^k; w^\dagger(x^k, y^k)) + c\mathcal{R}_y(x^k, y^k) \quad (20)$$

where $c$ is chosen to balance the coefficient of the term $\mathcal{R}_y(x^k, y^k)$ and $\|d_x^k\|^2$ such that

$$\mathbb{V}^{k+1} - \mathbb{V}^k \leq -\mathcal{O}(\alpha)\|d_x^k\|^2 - \mathcal{O}(\alpha)\mathcal{R}_y(x^k, y^k) + \mathcal{O}(\alpha)b_k^2. \quad (21)$$

Telescoping to (21) results in the convergence of both $\|d_x^k\|^2$ and $\mathcal{R}_y(x^k, y^k)$. By definition, $\|d_x^k\|^2 = \mathcal{R}_x(x^k, y^{k+1}, w^{k+1})$, so the convergence of $\mathcal{R}_x(x^k, y^k, w^k)$ is implied by the Lipschitz continuity of increments and the fact that $\|y^{k+1} - y^k\|, \|w^{k+1} - w^k\| \to 0$. Likewise, the convergence of $\mathcal{R}_w(x^k, y^k, w^k)$ can be established by recognizing that $b_k^2$ is sufficiently small when selecting a large $T$, ensuring that $\mathcal{R}_w(x^k, y^{k+1}, w^{k+1})$ is also small, and consequently, $\mathcal{R}_w(x^k, y^k, w^k)$. The convergence results of GALET can be formally stated as follows.

> **Theorem 2.** *Under Assumption 1–2, choosing $\alpha, \rho, \beta = \mathcal{O}(1)$ with some proper constants and $N = \mathcal{O}(1), T = \mathcal{O}(\log(K))$, then it holds that*
>
> $$\frac{1}{K}\sum_{k=0}^{K-1}\mathcal{R}_x(x^k, y^k, w^k) = \mathcal{O}\left(\frac{1}{K}\right), \quad \frac{1}{K}\sum_{k=0}^{K-1}\mathcal{R}_w(x^k, y^k, w^k) = \mathcal{O}\left(\frac{1}{K}\right), \quad \frac{1}{K}\sum_{k=0}^{K-1}\mathcal{R}_y(x^k, y^k) = \mathcal{O}\left(\frac{1}{K}\right).$$

From Definition 3, one need $\mathcal{O}(KT) = \mathcal{O}(\epsilon^{-1} \log(\epsilon^{-1})) =: \tilde{\mathcal{O}}(\epsilon^{-1})$ iterations in average to achieve $\epsilon$-stationary point of the BLO problem (1). The next remark shows that this complexity is optimal.

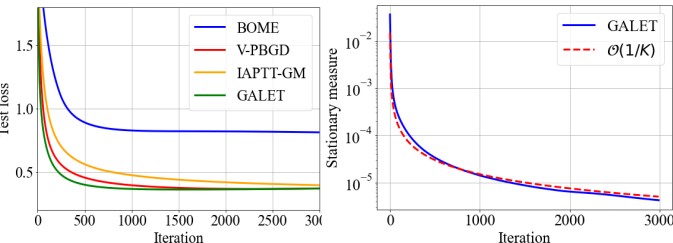

Figure 5: Convergence results of GALET of hyper-cleaning task on MNIST dataset. **Left**: test loss v.s. iteration. **Right**: stationary measure v.s. iteration (the $y$-axis is in the logarithmic scale).

**Remark 2** (**Lower bound**). The iteration complexity $\tilde{\mathcal{O}}(\epsilon^{-1})$ in Theorem 2 is optimal in terms of $\epsilon$, up to logarithmic factors. This is because when $f(x, y)$ is independent of $y$ and $g(x, y) = 0$, Assumptions 1 and 2 reduce to the smoothness assumption of $f$ on $x$, and thus the $\mathcal{O}(\epsilon^{-1})$ lower bound of nonconvex and smooth minimization in [6] also applies to BLO under Assumptions 1 and 2.

## 5 Preliminary Simulations and Conclusions

The goal of our simulations is to validate our theories and test the performance of GALET on actual learning applications. The experimental setting and parameter choices are included in the Appendix.

**Our stationary measure is a necessary condition of the global optimality.** As shown in Figure 1, GALET approaches the global optimal set of Example 1 and our stationary measure also converges to 0, while the value-function based KKT score does not.

**UL stepsize is the most sensitive parameter in our algorithm.** As shown in Figure 4, varying UL stepsize $\alpha$ leads to different trajectories of GALET, while varying parameters in the LL and shadow implicit gradient level only cause small perturbations. This means that $\alpha$ is the most sensitive parameter. The fact that GALET is robust to small values of $T$ and $N$ makes it computationally appealing for practical applications, eliminating the need to tune them extensively. This phenomenon also occurs in the hyper-cleaning task.

**Our method converges fast on the actual machine learning application.** We compare GALET with BOME [37], IAPTT-GM [43] and V-PBGD [57] in the hyper-cleaning task on the MNIST dataset. As shown in Figure 5, GALET converges faster than other methods and the convergence rate of GALET is $\mathcal{O}(1/K)$, which matches Theorem 2. Table 2 shows that the test accuracy of GALET is comparable to other methods.

**Conclusions and limitations.** In this paper, we study BLO with lower-level objectives satisfying the PL condition. We first establish the stationary measure for nonconvex-PL BLO without additional CQs and then propose an alternating optimization algorithm that generalizes the existing alternating (implicit) gradient-based algorithms for bilevel problems with a strongly convex lower-level objective. Our algorithm termed GALET achieves the $\tilde{\mathcal{O}}(\epsilon^{-1})$ iteration complexity for BLO with convex PL LL problems. Numerical experiments are provided to verify our theories.

| Method | Accuracy |
|---|---|
| BOME | 88.20 |
| IAPTT-GM | 90.13 |
| V-PBGD | 90.24 |
| **GALET** | 90.20 |

Table 2: Comparison of the test accuracy.

One potential limitation is that we only consider the deterministic BLO problem and require second-order information, and it is unclear whether our analysis can be generalized to handle the stochastic case and whether the same iteration complexity can be guaranteed with full first-order information. Another limitation is Assumption 2. This stipulates that the singular value of the LL Hessian must be bounded away from 0 to ensure the stability of the $\{w^k\}$.

## Acknowledgement

The work of Q. Xiao and T. Chen was supported by National Science Foundation MoDL-SCALE project 2134168, the RPI-IBM Artificial Intelligence Research Collaboration (AIRC) and an Amazon Research Award. We also thank anonymous reviewers and area chair for their insightful feedback and constructive suggestions, which are instrumental in enhancing the clarity and quality of our paper.

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

# Supplementary Material for "A Generalized Alternating Method for Bilevel Optimization under the Polyak-Łojasiewicz Condition"

## Table of Contents

## A   Auxiliary lemmas

We use $\| \cdot \|$ to be the Euclidean norm and $\mathcal{X}$ be the metric space with the Euclidean distance metric. Let $x \in \mathcal{X}$ be a point and $A \subset \mathcal{X}$ be a set, we define the distance of $x$ and $A$ as

$$d(x, A) = \inf\{\|x - a\| \mid a \in A\}.$$

The following lemma relates the PL condition to the error bound (EB) and the quadratic growth (QG) condition.

**Lemma 7** ([33, Theorem 2]). *If $g(x, y)$ is $\ell_{g,1}$-Lipschitz smooth and PL in $y$ with $\mu_g$, then it satisfies the EB condition with $\mu_g$, i.e.*

$$\|\nabla_y g(x, y)\| \geq \mu_g d(y, S(x)). \tag{22}$$

*Moreover, it also satisfies the QG condition with $\mu_g$, i.e.*

$$g(x, y) - g^*(x) \geq \frac{\mu_g}{2} d(y, S(x))^2. \tag{23}$$

*Conversely, if $g(x, y)$ is $\ell_{g,1}$-Lipschitz smooth and satisfies EB with $\mu_g$, then it is PL in $y$ with $\mu_g/\ell_{g,1}$.*

The following lemma shows that the calmness condition ensures the KKT conditions hold at the global minimizer.

**Proposition 1** ([11, Proposition 6.4.4]). *Let $(x^*, y^*)$ be the global minimzer of a constrained optimization problem and assume that this problem is calm at $(x^*, y^*)$. Then the KKT conditions hold at $(x^*, y^*)$.*

The following lemma gives the gradient of the value function.

**Lemma 8** ([51, Lemma A.5]). *Under Assumption 2 and assuming $g(x, y)$ is $\ell_{g,1}$- Lipschitz smooth, $g^*(x)$ is differentiable with the gradient*

$$\nabla g^*(x) = \nabla_x g(x, y), \;\; \forall y \in S(x).$$

*Moreover, $g^*(x)$ is $L_g$-smooth with $L_g := \ell_{g,1}(1 + \ell_{g,1}/2\mu_g)$.*

# B  Proof of Stationary Metric

## B.1  Theorem 3 and its proof

**Theorem 3.** *For nonconvex-PL BLO, A point $(x^*, y^*)$ is a local (resp. global) optimal solution of* (1) *if and only if it is a local (resp. global) optimal solution of* (3). *The same argument holds for* (2).

*Proof.* First, we prove the equivalence of (1) and reformulation (3). According to PL condition,

$$\frac{1}{2}\|\nabla_y g(x, y)\|^2 \geq \mu_g(g(x, y) - g^*(x)).$$

Therefore, if $\nabla_y g(x, y) = 0$, it holds that $0 \geq g(x, y) - g^*(x) \geq 0$ which yields $g(x, y) = g^*(x)$. On the other hand, if $g(x, y) = g^*(x)$, the first order necessary condition says $\nabla_y g(x, y) = 0$. As a consequence, we know that

$$\nabla_y g(x, y) = 0 \Leftrightarrow g(x, y) = g^*(x) \Leftrightarrow y \in S(x).$$

In this way, for any local (resp. global) optimal solution $(x^*, y^*)$ of (3), there exists a neighborhood $U$ such that for any $(x, y) \in U$ with $\nabla_y g(x, y) = 0$, it holds that $f(x^*, y^*) \leq f(x, y)$. Since $\nabla_y g(x, y) = 0 \Leftrightarrow y \in S(x)$, we know $(x^*, y^*)$ is also a local (resp. global) optimal solution of (2). The reverse holds true for the same reason.

The equivalence of Problem (1) and reformulation (2) can be proven in the same way. □

## B.2  Proof of Lemma 1

First, the gradient of $g(x, y)$ can be calculated by

$$\nabla_y g(x, y) = (x + y_1 - \sin(y_2))[1, -\cos(y_2)]^\top.$$

Since $\|[1, -\cos(y_2)]^\top\| \geq 1$ and $g^*(x) = 0$, we know

$$\frac{1}{2}\|\nabla_y g(x, y)\|^2 = \frac{1}{2}(x + y_1 - \sin(y_2))^2\|[1, -\cos(y_2)]^\top\|^2 \geq (g(x, y) - g^*(x)).$$

Thus $g(x, y)$ satisfies the PL condition with $\mu_g = 1$.

Next, the LL problem in Example 1 is equal to $x + y_1 - \sin(y_2) = 0$ so that the BLO of

$$\min_{x,y} f(x, y) = x^2 + y_1 - \sin(y_2), \quad \text{s.t. } y \in \arg\min g(x, y) = \frac{1}{2}(x + y_1 - \sin(y_2))^2 \tag{24}$$

is equivalent to

$$\min_{x,y} f(x, y) = x^2 + y_1 - \sin(y_2), \quad \text{s.t. } x + y_1 - \sin(y_2) = 0.$$

According to the constraint, we know $y_1 - \sin(y_2) = -x$ so that solving the BLO amounts to solving $\min\{x^2 - x\}$ with the constraint $x + y_1 - \sin(y_2) = 0$. Therefore, the global minimizers of BLO are those points $(x^*, y^*)$ where $x^* = 0.5$ and $y^* = (y_1^*, y_2^*)^\top$ with $y_1^* - \sin(y_2^*) + 0.5 = 0$.

On the other hand, reformulating (24) using (2) yields

$$\min_{x,y} f(x, y) = x^2 + y_1 - \sin(y_2), \quad \text{s.t. } g(x, y) - g^*(x) = 0 \tag{25}$$

If the calmness holds for (25), then according to Proposition 1, there exists $w^* \neq 0$ such that

$$\nabla_x f(x^*, y^*) + (\nabla_x g(x^*, y^*) - \nabla_x g^*(x^*))w^* = 0. \tag{26}$$

However, for any $w^* \neq 0$, $g^*(x) = 0$ and

$$\nabla_x f(x^*, y^*) + (\nabla_x g(x^*, y^*) - \nabla_x g^*(x^*))w^* = 2x^* + (x^* + y_1^* - \sin(y_2^*))w^* = 2x^* = 1 \neq 0$$

which contradicts (26). Therefore, the calmness condition fails for (24) with reformulation (2).

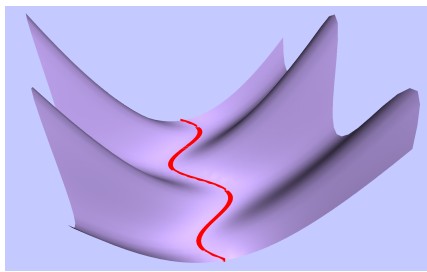

Figure 6: Landscape of $g(1, y)$ in Example 1 and the red line denotes $S(1)$. Different choices of $x$ only translate the plot and share the same landscape.

## B.3 Proof of Lemma 2

Figure 6 shows the landscape of $g(x, y)$ when $x = 1$ and the red line denotes its solution set. The figure reveals that $g(x, y)$ is nonconvex in $y$ with multiple LL solutions, and the solution set is closed but nonconvex.

*Proof.* It is easy to see that $S(x) = \{y \mid x + y_1 = \sin(y_2)\}$ and $g^*(x) = 0$.

First, the Slater condition is invalid because $\nabla_y g(x, y) = 0$ is an equality constraint.

Next, we can compute the Hessian and Jacobian of $g(x, y)$ as

$$\nabla^2_{yy} g(x, y) = \begin{bmatrix} 1 & -\cos(y_2) \\ -\cos(y_2) & \cos^2(y_2) + \sin(y_2)(x + y_1 - \sin(y_2)) \end{bmatrix}, \quad \nabla^2_{yx} g(x, y) = \begin{bmatrix} 1 \\ -\cos(y_2) \end{bmatrix}.$$

Since $\nabla^2_{yx} g(x, y)$ coincides with the first column of $\nabla^2_{yy} g(x, y)$, we know that

$$\text{rank}([\nabla^2_{yy} g(x, y), \quad \nabla^2_{yx} g(x, y)]) = \text{rank}(\nabla^2_{yy} g(x, y)).$$

For any global minimizer $(x^*, y^*)$ of the bilevel problem, it should satisfy the LL optimality, i.e. $y^* \in S(x^*)$, and thus $x^* + y_1^* - \sin(y_2^*) = 0$. In this way, we have

$$\nabla^2_{yy} g(x^*, y^*) = \begin{bmatrix} 1 & -\cos(y_2^*) \\ -\cos(y_2^*) & \cos^2(y_2^*) \end{bmatrix}.$$

Thus, $\text{rank}(\nabla^2_{yy} g(x^*, y^*)) = 1 \neq 2$ which contradicts the LICQ and MFCQ conditions.

Moreover, it follows that $\text{rank}(\nabla^2_{yy} g(x, y)) = 1$ if and only if

$$\det(\nabla^2_{yy} g(x, y)) = \sin(y_2)(x + y_1 - \sin(y_2)) = 0.$$

Thus, the CRCQ condition is equivalent to that there exists a neighborhood of $(x^*, y^*)$ such that either $\sin(y_2) = 0$ or $x + y_1 - \sin(y_2) = 0$ holds for any $(x, y)$ in that neighborhood. We fix $x^*$ and search such neighborhood for $y$ first. It is clear that finding an area of $y$ such that $\sin(y_2) \equiv 0$ is impossible due to the nature of sin function. On the other hand, $x^* + y_1 - \sin(y_2) = 0$ is equivalent to $y \in S(x^*)$ so that making $x^* + y_1 - \sin(y_2) \equiv 0$ in an neighborhood of $y^*$ is also unrealistic since otherwise, the solution set $S(x^*)$ should be open, which contradicts that $S(x)$ is closed. Thus, (6) violates CRCQ condition. $\square$

## B.4 Proof of Lemma 3

Assume the PL and the Lipschitz smoothness constants of $g(x, \cdot)$ are $\mu_g$ and $\ell_{g,1}$, and the Lipschitz continuity constant of $f(x, \cdot)$ is $\ell_{f,0}$. Consider $\forall q$, and $\forall (x', y')$, s.t. $\nabla_y g(x', y') + q = 0$, then letting $y_q \in \text{Proj}_{S(x')}(y')$ and according to Lemma 7, one has

$$\|q\| = \|\nabla_y g(x', y')\| \geq \mu_g \|y' - y_q\|$$

Since $(x^*, y^*)$ solves (3) and $(x', y_q)$ is also feasible to (3), one has $f(x^*, y^*) \leq f(x', y_q)$. Thus,

$$f(x', y') - f(x^*, y^*) \geq f(x', y') - f(x', y_q) \geq -\ell_{f,0} \|y' - y_q\| \geq -\frac{\ell_{f,0}}{\mu_g} \|q\|.$$

This justifies the calmness definition in (5) with $M := \frac{\ell_{f,0}}{\mu_g}$.

### B.5 Proof of Theorem 1

*Proof.* By applying Lemma 3 and Proposition 1, we know that KKT conditions are necessary, i.e.

$$\nabla_x f(x^*, y^*) + \nabla_{xy}^2 g(x^*, y^*) w^* = 0 \tag{27a}$$

$$\nabla_y f(x^*, y^*) + \nabla_{yy}^2 g(x^*, y^*) w^* = 0 \tag{27b}$$

$$\nabla_y g(x^*, y^*) = 0. \tag{27c}$$

On the other hand, since

$$\nabla_y g(x^*, y^*) = 0 \Leftrightarrow g(x^*, y^*) - g^*(x^*) = 0$$

$$\nabla_y f(x^*, y^*) + \nabla_{yy}^2 g(x^*, y^*) w^* = 0 \Rightarrow \nabla_{yy}^2 g(x, y) \left( \nabla_y f(x^*, y^*) + \nabla_{yy}^2 g(x^*, y^*) w^* \right) = 0$$

we arrive at the conclusion. □

## C Proof of Lower-level Error

In this section, we give the proof of Lemma 4. To prove (17a) in Lemma 4, by the smoothness of $g(x, y)$ and the update rule, we have

$$g(x^k, y^{k,n+1}) \le g(x^k, y^{k,n}) + \langle \nabla_y g(x^k, y^{k,n}), y^{k,n+1} - y^{k,n} \rangle + \frac{\ell_{g,1}}{2} \|y^{k,n+1} - y^{k,n}\|^2$$

$$\le g(x^k, y^{k,n}) - \beta \|\nabla_y g(x^k, y^{k,n})\|^2 + \frac{\beta^2 \ell_{g,1}}{2} \|\nabla_y g(x^k, y^{k,n})\|^2$$

$$= g(x^k, y^{k,n}) - \left( \beta - \frac{\beta^2 \ell_{g,1}}{2} \right) \|\nabla_y g(x^k, y^{k,n})\|^2$$

$$\overset{(a)}{\le} g(x^k, y^{k,n}) - \left( 2\beta - \beta^2 \ell_{g,1} \right) \mu_g \left( g(x^k, y^{k,n}) - g^*(x^k) \right) \tag{28}$$

where (a) results from Assumption 2. Then subtracting $g^*(x^k)$ from both sides of (28), we get

$$g(x^k, y^{k,n+1}) - g^*(x^k) \le \left[ 1 - (2\beta - \beta^2 \ell_{g,1}) \mu_g \right] \left( g(x^k, y^{k,n}) - g^*(x^k) \right).$$

By plugging in $\beta \le \frac{1}{\ell_{g,1}}$ and applying $N$ times, we get the conclusion in (17a).

To prove (17b), we decompose the error by

$$g(x^{k+1}, y^{k+1}) - g^*(x^{k+1}) = \left( g(x^k, y^{k+1}) - g^*(x^k) \right)$$

$$+ \underbrace{\left[ \left( g(x^{k+1}, y^{k+1}) - g^*(x^{k+1}) \right) - \left( g(x^k, y^{k+1}) - g^*(x^k) \right) \right]}_{J_1}. \tag{29}$$

To obtain the upper bound of $J_1$, we first notice that $g(x, y^{k+1}) - g^*(x)$ is $(\ell_{g,1} + L_g)$-Lipschitz smooth over $x$ according to the smoothness of $g(x, y^{k+1})$ in Assumption 1 and $g^*(x)$ from Lemma 8. Therefore, we have

$$J_1 \le \langle \nabla_x g(x^k, y^{k+1}) - \nabla g^*(x^k), x^{k+1} - x^k \rangle + \frac{\ell_{g,1} + L_g}{2} \|x^{k+1} - x^k\|^2$$

$$\le -\alpha \langle \nabla_x g(x^k, y^{k+1}) - \nabla g^*(x^k), d_x^k \rangle + \frac{\ell_{g,1} + L_g}{2} \alpha^2 \|d_x^k\|^2$$

$$\le \alpha \|\nabla_x g(x^k, y^{k+1}) - \nabla g^*(x^k)\| \|d_x^k\| + \frac{\ell_{g,1} + L_g}{2} \alpha^2 \|d_x^k\|^2$$

$$\overset{(a)}{\le} \alpha \ell_{g,1} d(y^{k+1}, S(x^k)) \|d_x^k\| + \frac{\ell_{g,1} + L_g}{2} \alpha^2 \|d_x^k\|^2$$

$$\overset{(b)}{\le} \frac{\alpha \eta_1 \ell_{g,1}}{2} d(y^{k+1}, S(x^k))^2 + \frac{\alpha \ell_{g,1}}{2\eta_1} \|d_x^k\|^2 + \frac{\ell_{g,1} + L_g}{2} \alpha^2 \|d_x^k\|^2$$

$$\overset{(c)}{\le} \frac{\alpha \eta_1 \ell_{g,1}}{\mu_g} (g(x^k, y^{k+1}) - g^*(x^k)) + \frac{\alpha \ell_{g,1}}{2\eta_1} \|d_x^k\|^2 + \frac{\ell_{g,1} + L_g}{2} \alpha^2 \|d_x^k\|^2 \tag{30}$$

where (a) is derived from Lipschitz continuity of $\nabla g(x, y)$ and Lemma 8, (b) comes from Cauchy-Swartz inequality, (c) is due to Lemma 7. Plugging (30) into (29) yields the conclusion in (17b).

# D Proof of Shadow Implicit Gradient Error

We first present some properties of $\mathcal{L}(x, y, w)$ and then prove Lemma 5. We define

$$\mathcal{L}^*(x, y) = \min_w \mathcal{L}(x, y, w), \quad \mathcal{W}(x, y) = \arg\min_w \mathcal{L}(x, y, w)$$

## D.1 Propositions of $\mathcal{L}(x, y, w)$ and $\mathcal{L}^*(x, y)$

**Proposition 2.** *Under Assumption 1–2, $\mathcal{L}(x, y, w)$ is $\sigma_g^2$-PL and $\ell_{g,1}^2$- Lipschitz smooth over $w$.*

*Proof.* According to the definition of $\mathcal{L}(x, y, w)$, we can calculate the Hessian of it as

$$\nabla_{ww}^2 \mathcal{L}(x, y, w) = \nabla_{yy}^2 g(x, y) \nabla_{yy}^2 g(x, y).$$

Since $\nabla_{ww} \mathcal{L}(x, y, w) \preceq \ell_{g,1}^2 I$, we know that $\mathcal{L}(x, y.w)$ is $\ell_{g,1}^2$- Lipschitz smooth with respect to $w$. On the other hand, $\mathcal{L}(x, y, w)$ is $\sigma_g^2$-PL since $\frac{1}{2}\|Aw + b\|^2$ is a strongly convex function composited with a linear function which is $\sigma_{\min}^2(A)$-PL according to [33]. $\qquad\square$

## D.2 Proof of Lemma 5

According to Proposition 2 and following the same proof of Lemma (17a), we know that

$$\mathcal{L}(x^k, y^{k+1}, w^{k+1}) - \mathcal{L}^*(x^k, y^{k+1}) \le (1 - \rho\sigma_g^2)^T \left( \mathcal{L}(x^k, y^{k+1}, w^{k,0}) - \mathcal{L}^*(x^k, y^{k+1}) \right)$$
$$\le (1 - \rho\sigma_g^2)^T \ell_{f,0}^2$$

where the last inequality results from $\mathcal{L}(x^k, y^{k+1}, w^0) = \|\nabla_y f(x^k, y^{k+1})\|^2/2 \le \ell_{f,0}/2$ and $\mathcal{L}^*(x^k, y^{k+1}) \ge 0$. Then according to the EB condition, we know

$$d(w^{k+1}, \mathcal{W}(x^k, y^{k+1}))^2 \le (1 - \rho\sigma_g^2)^T \frac{2\ell_{f,0}^2}{\mu_g}$$

On the other hand, since $w^{k,t} \in \text{Ran}(\nabla_{yy}^2 g(x^k, y^{k+1}))$ according to the update, then $w^{k+1} \in \text{Ran}(\nabla_{yy}^2 g(x^k, y^{k+1}))$ and thus

$$\arg\min_{w \in \mathcal{W}(x^k, y^{k+1})} \|w - w^{k+1}\| = w^\dagger(x^k, y^{k+1}).$$

As a result, the bias can be bounded by

$$b_k^2 = \|w^{k+1} - w^\dagger(x^k, y^{k+1})\|^2 = d(w^{k+1}, \mathcal{W}(x^k, y^{k+1}))^2 \le (1 - \rho\sigma_g^2)^T \frac{2\ell_{f,0}^2}{\mu_g}.$$

# E Proof of Upper-level Error

## E.1 Propositions of $w^\dagger(x, y)$

Before proving the UL descent, we first present some property of $w^\dagger(x, y)$.

**Lemma 9** (Lipschitz continuity and boundedness of $w^\dagger(x, y)$)**.** *Under Assumption 1–2, for any $x, x_1, x_2$ and $y, y_1, y_2$, it holds that*

$$\|w^\dagger(x, y)\| \le \ell_{f,0}/\sigma_g, \quad \|w^\dagger(x_1, y_1) - w^\dagger(x_2, y_2)\| \le L_w \|(x_1, y_1) - (x_2, y_2)\|$$
$$\text{with } L_w = \frac{\ell_{f,1}}{\sigma_g} + \frac{\sqrt{2}\ell_{g,2}\ell_{f,0}}{\sigma_g^2}.$$

*Proof.* First, $\left(\nabla_{yy}^2 g(x, y)\right)^\dagger$ is bounded since according to Assumption 2,

$$\| \left(\nabla_{yy}^2 g(x, y)\right)^\dagger \| \le \frac{1}{\sigma_{\min}(\nabla_{yy}^2 g(x, y))} \le \frac{1}{\sigma_g}.$$

On the other hand, according to [60, Theorem 3.3],

$$
\| \left(\nabla_{yy}^2 g(x,y)\right)^\dagger - \left(\nabla_{yy}^2 g(x',y')\right)^\dagger \|
$$
$$
\leq \sqrt{2} \max\{\| \left(\nabla_{yy}^2 g(x,y)\right)^\dagger \|^2, \| \left(\nabla_{yy}^2 g(x',y')\right)^\dagger \|^2\} \|\nabla_{yy}^2 g(x,y) - \nabla_{yy}^2 g(x',y')\|
$$
$$
\leq \frac{\sqrt{2}\ell_{g,2}}{\sigma_g^2} \|(x,y) - (x',y')\|
$$

which says $\left(\nabla_{yy}^2 g(x,y)\right)^\dagger$ is $(\sqrt{2}\ell_{g,2}/\sigma_g^2)$- Lipschitz continuous.

By the definition (32), $w^\dagger(x,y) = -\left(\nabla_{yy}^2 g(x,y)\right)^\dagger \nabla_y f(x,y)$. Therefore, the boundedness of $w^\dagger(x,y)$ is given by

$$
\|w^\dagger(x,y)\| \leq \| \left(\nabla_{yy}^2 g(x,y)\right)^\dagger \|\|\nabla_y f(x,y)\| \leq \frac{\ell_{f,0}}{\sigma_g}.
$$

Besides, for any $x_1, x_2$ and $y_1, y_2$, we have

$$
\|w^\dagger(x_1,y_1) - w^\dagger(x_2,y_2)\|
$$
$$
= \| - \left(\nabla_{yy}^2 g(x_1,y_2)\right)^\dagger \nabla_y f(x_1,y_1) + \left(\nabla_{yy}^2 g(x_2,y_2)\right)^\dagger \nabla_y f(x_2,y_2)\|
$$
$$
\overset{(a)}{\leq} \| \left(\nabla_{yy}^2 g(x_1,y_1)\right)^\dagger \|\|\nabla_y f(x_1,y_1) - \nabla_y f(x_2,y_2)\|
$$
$$
+ \|\nabla_y f(x_2,y_2)\|\| \left(\nabla_{yy}^2 g(x_1,y_1)\right)^\dagger - \left(\nabla_{yy}^2 g(x_2,y_2)\right)^\dagger \|
$$
$$
\overset{(b)}{\leq} \left( \frac{\ell_{f,1}}{\sigma_g} + \frac{\sqrt{2}\ell_{g,2}\ell_{f,0}}{\sigma_g^2} \right) \|(x_1,y_1) - (x_2,y_2)\|
$$

where (a) is due to $C_1 D_1 - C_2 D_2 = C_1(D_1 - D_2) + D_2(C_1 - C_2)$ and Cauchy-Schwartz inequality and (b) comes from Lemma 9. $\qquad\square$

## E.2 Proof of Lemma 6

We aim to prove the descent of upper-level by

$$
F(x^{k+1}, y^{k+1}; w^\dagger(x^{k+1}, y^{k+1})) - F(x^k, y^k; w^\dagger(x^k, y^k))
$$
$$
= \underbrace{F(x^{k+1}, y^{k+1}; w^\dagger(x^{k+1}, y^{k+1})) - F(x^{k+1}, y^{k+1}; w^\dagger(x^k, y^{k+1}))}_{\text{Lemma 11}}
$$
$$
+ \underbrace{F(x^{k+1}, y^{k+1}; w^\dagger(x^k, y^{k+1})) - F(x^k, y^{k+1}; w^\dagger(x^k, y^{k+1}))}_{\text{Lemma 12}}
$$
$$
+ \underbrace{F(x^k, y^{k+1}; w^\dagger(x^k, y^{k+1})) - F(x^k, y^{k+1}; w^\dagger(x^k, y^k))}_{\text{Lemma 13}}
$$
$$
+ \underbrace{F(x^k, y^{k+1}; w^\dagger(x^k, y^k)) - F(x^k, y^k; w^\dagger(x^k, y^k))}_{\text{Lemma 14}}. \tag{31}
$$

For ease of use, we derive the gradient of $F(x,y; w^\dagger(x,y))$ below.

$$
\nabla_x F(x,y; w^\dagger(x,y)) = \nabla_x f(x,y) + \nabla_{xy}^2 g(x,y) w^\dagger(x,y) \tag{32a}
$$
$$
\nabla_y F(x,y; w^\dagger(x,y)) = \nabla_y f(x,y) + \nabla_{yy}^2 g(x,y) w^\dagger(x,y) \tag{32b}
$$

**Lemma 10.** *Under Assumption 1–2, $F(x,y;w)$ is $(\ell_{f,1} + \ell_{g,2}\|w\|)$-smooth over $x$ and $y$.*

*Proof.* This is implied by the Lipschitz smoothness of $f(x,y)$ and $\nabla_y g(x,y)$. $\qquad\square$

**Lemma 11.** *Under Assumption 1–2, we have*

$$
F(x^{k+1}, y^{k+1}; w^\dagger(x^{k+1}, y^{k+1})) - F(x^{k+1}, y^{k+1}; w^\dagger(x^k, y^{k+1}))
$$
$$
\leq \frac{\beta\eta_2\ell_{g,1}^2}{\mu_g} \mathcal{R}_y(x^k, y^k) + \left( \frac{\alpha^2 L_w^2}{2\beta\eta_2} + \ell_{g,1}L_w\alpha^2 \right) \|d_x^k\|^2 \tag{33}
$$

*where $\eta_2$ is a constant that will be chosen in Theorem 2.*

*Proof.* According to the definition of $F(x,y) = f(x,y) + w^\top \nabla_y g(x,y)$, it holds that

$$F(x^{k+1}, y^{k+1}; w^\dagger(x^{k+1}, y^{k+1})) - F(x^{k+1}, y^{k+1}; w^\dagger(x^k, y^{k+1}))$$

$$= f(x^{k+1}, y^{k+1}) + \langle \nabla_y g(x^{k+1}, y^{k+1}), w^\dagger(x^{k+1}, y^{k+1}) \rangle - f(x^{k+1}, y^{k+1}) - \langle \nabla_y g(x^{k+1}, y^{k+1}), w^\dagger(x^k, y^{k+1}) \rangle$$

$$= \langle \nabla_y g(x^{k+1}, y^{k+1}), w^\dagger(x^{k+1}, y^{k+1}) - w^\dagger(x^k, y^{k+1}) \rangle$$

$$\leq \langle \nabla_y g(x^k, y^{k+1}), w^\dagger(x^{k+1}, y^{k+1}) - w^\dagger(x^k, y^{k+1}) \rangle$$

$$\quad + \langle \nabla_y g(x^{k+1}, y^{k+1}) - \nabla_y g(x^k, y^{k+1}), w^\dagger(x^{k+1}, y^{k+1}) - w^\dagger(x^k, y^{k+1}) \rangle$$

$$\overset{(a)}{\leq} \frac{\beta \eta_2}{2} \|\nabla_y g(x^k, y^{k+1})\|^2 + \frac{\alpha^2 L_w^2}{2\beta \eta_2} \|d_x^k\|^2 + \ell_{g,1} L_w \alpha^2 \|d_x^k\|^2$$

$$\overset{(b)}{\leq} \frac{\beta \eta_2 \ell_{g,1}^2}{\mu_g} \mathcal{R}_y(x^k, y^k) + \left( \frac{\alpha^2 L_w^2}{2\beta \eta_2} + \ell_{g,1} L_w \alpha^2 \right) \|d_x^k\|^2$$

where (a) is earned by Young's inequality and the Lipschitz continuity of $w^\dagger(x,y)$ and $\nabla_y g$, (b) is resulting from letting $y^* = \arg\min_{y \in S(x^k)} \|y - y^k\|$ and Lemma 7, that is

$$\|\nabla_y g(x^k, y^{k+1})\|^2 = \|\nabla_y g(x^k, y^{k+1}) - \nabla_y g(x^k, y^*)\|^2$$

$$\leq \ell_{g,1}^2 \|y^{k+1} - y^*\|^2 = \ell_{g,1}^2 d(y^{k+1}, S(x^k))^2$$

$$\leq \frac{2\ell_{g,1}^2}{\mu_g} \left( g(x^k, y^{k+1}) - g^*(x^k) \right)$$

$$\leq \frac{2\ell_{g,1}^2}{\mu_g} \left( g(x^k, y^k) - g^*(x^k) \right) = \frac{2\ell_{g,1}^2}{\mu_g} \mathcal{R}_y(x^k, y^k). \tag{34}$$

$\square$

**Lemma 12.** *Under Assumption 1–2, we have*

$$F(x^{k+1}, y^{k+1}; w^\dagger(x^k, y^{k+1})) - F(x^k, y^{k+1}; w^\dagger(x^k, y^{k+1}))$$

$$\leq - \left( \frac{\alpha}{2} - \frac{L_F}{2} \alpha^2 \right) \|d_x^k\|^2 + \frac{\alpha \ell_{g,1}^2}{2} b_k^2 \tag{35}$$

*where $b_k := \|w^{k+1} - w^\dagger(x^k, y^{k+1})\|$, and the constant $L_F$ is defined as $L_F = \ell_{f,0}(\ell_{f,1} + \ell_{g,2})/\sigma_g$.*

*Proof.* Since $F(x,y;w)$ is $(\ell_{f,1} + \ell_{g,2}\|w\|)$ Lipschitz smooth with respect to $x$, then according to Lemma 9, we have

$$F(x^{k+1}, y^{k+1}; w^\dagger(x^k, y^{k+1}))$$

$$\leq F(x^k, y^{k+1}; w^\dagger(x^k, y^{k+1})) + \langle \nabla_x f(x^k, y^{k+1}) + \nabla_{xy}^2 g(x^k, y^{k+1}) w^\dagger(x^k, y^{k+1}), x^{k+1} - x^k \rangle$$

$$\quad + \frac{L_F}{2} \|x^{k+1} - x^k\|^2$$

$$= F(x^k, y^{k+1}; w^\dagger(x^k, y^{k+1})) - \alpha \langle \nabla_x f(x^k, y^{k+1}) + \nabla_{xy}^2 g(x^k, y^{k+1}) w^\dagger(x^k, y^{k+1}), d_x^k \rangle + \frac{\alpha^2 L_F}{2} \|d_x^k\|^2$$

$$\overset{(a)}{=} F(x^k, y^{k+1}; w^\dagger(x^k, y^{k+1})) - \frac{\alpha}{2} \|d_x^k\|^2 - \frac{\alpha}{2} \|\nabla_x f(x^k, y^{k+1}) + \nabla_{xy}^2 g(x^k, y^{k+1}) w^\dagger(x^k, y^{k+1})\|^2$$

$$\quad + \frac{\alpha}{2} \|d_x^k - \nabla_x f(x^k, y^{k+1}) - \nabla_{xy}^2 g(x^k, y^{k+1}) w^\dagger(x^k, y^{k+1})\|^2 + \frac{\alpha^2 L_F}{2} \|d_x^k\|^2$$

$$\overset{(b)}{\leq} F(x^k, y^{k+1}; w^\dagger(x^k, y^{k+1})) - \left( \frac{\alpha}{2} - \frac{L_F}{2} \alpha^2 \right) \|d_x^k\|^2 + \frac{\alpha \ell_{g,1}^2}{2} b_k^2$$

where (a) is due to $\langle A, B \rangle = \frac{\|A\|^2}{2} + \frac{\|B\|^2}{2} - \frac{\|A-B\|^2}{2}$, (b) results from

$$\|d_x^k - \nabla_x f(x^k, y^{k+1}) - \nabla_{xy}^2 g(x^k, y^{k+1}) w^\dagger(x^k, y^{k+1})\|$$

$$= \|\nabla_x f(x^k, y^{k+1}) + \nabla_{yy}^2 g(x^k, y^{k+1}) w^{k+1} - \nabla_x f(x^k, y^{k+1}) - \nabla_{yy}^2 g(x^k, y^{k+1}) w^\dagger(x^k, y^{k+1})\|$$

$$= \|\nabla_{yy}^2 g(x^k, y^{k+1})(w^{k+1} - w^\dagger(x^k, y^{k+1}))\|$$

$$\leq \|\nabla_{yy}^2 g(x^k, y^{k+1})\|\|w^{k+1} - w^\dagger(x^k, y^{k+1})\|$$
$$\leq \ell_{g,1}\|w^{k+1} - w^\dagger(x^k, y^{k+1})\| = \ell_{g,1} b_k.$$

Rearranging terms yields the conclusion. $\qquad\square$

**Lemma 13.** *Under Assumption 1–2, we have*

$$F(x^k, y^{k+1}; w^\dagger(x^k, y^{k+1})) - F(x^k, y^{k+1}; w^\dagger(x^k, y^k)) \leq \frac{2\beta N L_w \ell_{g,1}^2}{\mu_g} \mathcal{R}_y(x^k, y^k) \qquad (36)$$

*Proof.* Letting $y_i^* = \mathrm{Proj}_{S(x^k)}(y^{k,i})$, we first bound $\|y^{k+1} - y^k\|^2$ by

$$\|y^{k+1} - y^k\|^2 = \|y^{k,N} - y^{k,N-1} + y^{k,N-1} - y^{k,N-2} + \cdots - y^{k,0}\|^2$$

$$\leq N \sum_{i=0}^{N-1} \|y^{k,i+1} - y^{k,i}\|^2 = \beta^2 N \sum_{i=0}^{N-1} \|\nabla_y g(x^k, y^{k,i})\|^2$$

$$\overset{(a)}{=} \beta^2 N \sum_{i=0}^{N-1} \|\nabla_y g(x^k, y^{k,i}) - \nabla_y g(x^k, y_i^*)\|^2$$

$$\leq \beta^2 N \ell_{g,1}^2 \sum_{i=0}^{N-1} \|y^{k,i} - y_i^*\|^2 = \beta^2 N \ell_{g,1}^2 \sum_{i=0}^{N-1} d(y^{k,i}, S(x^k))^2$$

$$\overset{(b)}{\leq} \frac{2\beta^2 N \ell_{g,1}^2}{\mu_g} \sum_{i=0}^{N-1} \left(g(x^k, y^{k,i}) - g^*(x^k)\right)$$

$$\overset{(c)}{\leq} \frac{2\beta^2 N \ell_{g,1}^2}{\mu_g} \sum_{i=0}^{N-1} (1 - \beta\mu_g)^i \left(g(x^k, y^k) - g^*(x^k)\right)$$

$$\leq \frac{2\beta^2 N^2 \ell_{g,1}^2}{\mu_g} \left(g(x^k, y^k) - g^*(x^k)\right) = \frac{2\beta^2 N^2 \ell_{g,1}^2}{\mu_g} \mathcal{R}_y(x^k, y^k). \qquad (37)$$

where (a) comes from $\nabla_y g(x^k, y_i^*) = 0$, (b) is due to quadratic growth condition in Lemma 7, (c) is derived from the contraction in (17a).

Then, according to the definition of $F(x, y; w) = f(x, y) + w^\top \nabla_y g(x, y)$, it holds that

$$F(x^k, y^{k+1}; w^\dagger(x^k, y^{k+1})) - F(x^k, y^{k+1}; w^\dagger(x^k, y^k))$$
$$= f(x^k, y^{k+1}) + \langle \nabla_y g(x^k, y^{k+1}), w^\dagger(x^k, y^{k+1}) \rangle - f(x^k, y^{k+1}) - \langle \nabla_y g(x^k, y^{k+1}), w^\dagger(x^k, y^k) \rangle$$
$$= \langle \nabla_y g(x^k, y^{k+1}), w^\dagger(x^k, y^{k+1}) - w^\dagger(x^k, y^k) \rangle$$
$$\leq \frac{\beta N L_w}{2} \|\nabla_y g(x^k, y^{k+1})\|^2 + \frac{L_w}{2\beta N} \|y^{k+1} - y^k\|^2$$
$$\overset{(a)}{\leq} \frac{2\beta N L_w \ell_{g,1}^2}{\mu_g} \mathcal{R}_y(x^k, y^k)$$

where (a) is resulting from (34) and (37).

$\qquad\square$

**Lemma 14.** *Under Assumption 1–2, we have*

$$F(x^k, y^{k+1}; w^\dagger(x^k, y^k)) - F(x^k, y^k; w^\dagger(x^k, y^k)) \leq \frac{2\beta\ell_{f,0}\ell_{g,2} + L_F \ell_{g,1}^2 \beta^2}{\mu_g} \mathcal{R}_y(x^k, y^k) \quad (38)$$

*Proof.* We can expand $F(x, y; w)$ with respect to $y$ by

$$F(x^k, y^{k+1}; w^\dagger(x^k, y^k))$$
$$\leq F(x^k, y^k; w^\dagger(x^k, y^k)) + \langle \nabla_y f(x^k, y^k) + \nabla_{yy}^2 g(x^k, y^k) w^\dagger(x^k, y^k), y^{k+1} - y^k \rangle + \frac{L_F}{2} \|y^{k+1} - y^k\|^2$$

$$= F(x^k, y^k; w^\dagger(x^k, y^k)) - \beta \langle \nabla_y f(x^k, y^k) + \nabla_{yy}^2 g(x^k, y^k) w^\dagger(x^k, y^k), \nabla_y g(x^k, y^k) \rangle + \frac{L_F}{2} \beta^2 \|\nabla_y g(x^k, y^k)\|^2$$

$$\overset{(a)}{\leq} F(x^k, y^k; w^\dagger(x^k, y^k)) + \frac{L_F \ell_{g,1}^2 \beta^2}{\mu_g} \mathcal{R}_y(x^k, y^k) \underbrace{- \beta \langle \nabla_y f(x^k, y^k) + \nabla_{yy}^2 g(x^k, y^k) w^\dagger(x^k, y^k), \nabla_y g(x^k, y^k) \rangle}_{J_2}$$

where (a) results from (37). Letting $y^* = \arg\min_{y \in S(x^k)} \|y - y^k\|$, the bound of $J_2$ is derived by

$$J_2 \overset{(a)}{=} -\beta \langle \nabla_y f(x^k, y^k) + \nabla_{yy}^2 g(x^k, y^k) w^\dagger(x^k, y^k), \nabla_y g(x^k, y^k) - \nabla_y g(x^k, y^*) \rangle$$

$$\overset{(b)}{=} -\beta \langle \nabla_y f(x^k, y^k) + \nabla_{yy}^2 g(x^k, y^k) w^\dagger(x^k, y^k), \nabla_{yy}^2 g(x^k, y')(y^k - y^*) \rangle$$

$$= -\beta \langle \nabla_y f(x^k, y^k) + \nabla_{yy}^2 g(x^k, y^k) w^\dagger(x^k, y^k), \nabla_{yy}^2 g(x^k, y^k)(y^k - y^*) \rangle$$

$$\quad + \beta \langle \nabla_y f(x^k, y^k) + \nabla_{yy}^2 g(x^k, y^k) w^\dagger(x^k, y^k), \left(\nabla_{yy}^2 g(x^k, y^k) - \nabla_{yy}^2 g(x^k, y')\right)(y^k - y^*) \rangle$$

$$\leq -\beta \langle \nabla_{yy}^2 g(x^k, y^k) \left(\nabla_y f(x^k, y^k) + \nabla_{yy}^2 g(x^k, y^k) w^\dagger(x^k, y^k)\right), y^k - y^* \rangle$$

$$\quad + \beta \|\nabla_y f(x^k, y^k) + \nabla_{yy}^2 g(x^k, y^k) w^\dagger(x^k, y^k)\| \|\nabla_{yy}^2 g(x^k, y^k) - \nabla_{yy}^2 g(x^k, y')\| \|y^k - y^*\|$$

$$\overset{(c)}{\leq} -\beta \langle \nabla_{yy}^2 g(x^k, y^k) \left(\nabla_y f(x^k, y^k) + \nabla_{yy}^2 g(x^k, y^k) w^\dagger(x^k, y^k)\right), y^k - y^* \rangle + \beta \ell_{f,0} \ell_{g,2} \|y^k - y^*\|^2$$

$$\overset{(d)}{\leq} \beta \ell_{f,0} \ell_{g,2} \|y^k - y^*\|^2 = \beta \ell_{f,0} \ell_{g,2} d(y^k, S(x^k))^2$$

$$\leq \frac{2\beta \ell_{f,0} \ell_{g,2}}{\mu_g} \left(g(x^k, y^k) - g^*(x^k)\right) = \frac{2\beta \ell_{f,0} \ell_{g,2}}{\mu_g} \mathcal{R}_y(x^k, y^k) \tag{39}$$

where (a) is due to $\nabla_y g(x^k, y^*) = 0$, (b) is due to the mean value theorem and there exists $t \in [0, 1]$ such that $y' = ty^k + (1-t)y^*$, (c) is due to $\|I - AA^\dagger\| \leq 1$ and

$$\|\nabla_y f(x^k, y^k) + \nabla_{yy}^2 g(x^k, y^k) w^\dagger(x^k, y^k)\| = \|(I - \nabla_{yy}^2 g(x^k, y^k) \left(\nabla_{yy}^2 g(x^k, y^k)\right)^\dagger \nabla_y f(x^k, y^k)\|$$

$$\leq \|\nabla_y f(x^k, y^k)\| \leq \ell_{f,0}$$

and (d) comes from $w^\dagger(x, y) = -\left(\nabla_{yy}^2 g(x^k, y^k)\right)^\dagger \nabla_y f(x, y)$ such that

$$\nabla_{yy}^2 g(x^k, y^k) \left(\nabla_y f(x^k, y^k) + \nabla_{yy}^2 g(x^k, y^k) w^\dagger(x^k, y^k)\right)$$

$$= \nabla_{yy}^2 g(x^k, y^k) \underbrace{\left(I - \nabla_{yy}^2 g(x^k, y^k) \left(\nabla_{yy}^2 g(x^k, y^k)\right)^\dagger\right) \nabla_y f(x^k, y^k)}_{\in \mathrm{Ker}(\nabla_{yy}^2 g(x^k, y^k))} = 0.$$

Here, $I - \nabla_{yy}^2 g(x^k, y^k) \left(\nabla_{yy}^2 g(x^k, y^k)\right)^\dagger$ is the orthogonal projector onto $\mathrm{Ker}(\nabla_{yy}^2 g(x^k, y^k))$. $\quad\square$

Finally, according to (31), Lemma 6 can be proved in the following way.

*Proof.* Adding (33), (35), (36) and (38), we get

$$F(x^{k+1}, y^{k+1}; w^\dagger(x^{k+1}, y^{k+1})) - F(x^k, y^k; w^\dagger(x^k, y^k))$$

$$\leq -\left(\frac{\alpha}{2} - \frac{L_w^2}{2\eta_2} \frac{\alpha^2}{\beta} - \frac{L_F}{2} \alpha^2 - \ell_{g,1} L_w \alpha^2\right) \|d_x^k\|^2 + \frac{\alpha \ell_{g,1}^2}{2} b_k^2$$

$$+ \frac{\beta \eta_2 \ell_{g,1}^2 + 2\beta N L_w \ell_{g,1}^2 + 2\beta \ell_{f,0} \ell_{g,2} + L_F \ell_{g,1}^2 \beta^2}{\mu_g} \mathcal{R}_y(x^k, y^k). \tag{40}$$

Then by choosing $\alpha \leq \min\left\{\frac{1}{4L_F + 8L_w \ell_{g,1}}, \frac{\beta \eta_2}{2L_w^2}\right\}$, we yield the conclusion. $\quad\square$

## F Proof of Overall Descent of Lyapunov Function

*Proof.* According to (17a), (17b) and (29), we know

$$\mathcal{R}_y(x^{k+1}, y^{k+1}) - \mathcal{R}_y(x^k, y^k) \leq \left(-\beta \mu_g + \frac{\alpha \eta_1 \ell_{g,1}}{\mu_g}\right) \mathcal{R}_y(x^k, y^k) + \left(\frac{\alpha \ell_{g,1}}{2\eta_1} + \frac{\ell_{g,1} + L_g}{2} \alpha^2\right) \|d_x^k\|^2 \tag{41}$$

Then since $\mathbb{V}^k = F(x^k, y^k; w^\dagger(x^k, y^k)) + c\mathcal{R}_y(x^k, y^k)$, then adding (40) and (41) yields

$$
\begin{aligned}
\mathbb{V}^{k+1} - \mathbb{V}^k \leq & -\left(\frac{\alpha}{2} - \frac{L_w^2}{2\eta_2}\frac{\alpha^2}{\beta} - \frac{L_F}{2}\alpha^2 - \ell_{g,1}L_w\alpha^2 - \frac{c\alpha\ell_{g,1}}{2\eta_1} - \frac{c(\ell_{g,1}+L_g)}{2}\alpha^2\right)\|d_x^k\|^2 + \frac{\alpha\ell_{g,1}^2}{2}b_k^2 \\
& -\left(c\beta\mu_g - \frac{c\alpha\eta_1\ell_{g,1} + \beta\eta_2\ell_{g,1}^2 + 2\beta NL_w\ell_{g,1}^2 + 2\beta\ell_{f,0}\ell_{g,2} + L_F\ell_{g,1}^2\beta^2}{\mu_g}\right)\mathcal{R}_y(x^k, y^k)
\end{aligned}
$$

$$
\begin{aligned}
\overset{(a)}{=} & -\left(\frac{\alpha}{4} - \frac{L_w^2}{2\eta_2}\frac{\alpha^2}{\beta} - \frac{L_F}{2}\alpha^2 - \ell_{g,1}L_w\alpha^2 - \frac{c(\ell_{g,1}+L_g)}{2}\alpha^2\right)\|d_x^k\|^2 + \frac{\alpha\ell_{g,1}^2}{2}b_k^2 \\
& -\left(c\beta\mu_g - \frac{2c^2\alpha\ell_{g,1}^2 + \beta\eta_2\ell_{g,1}^2 + 2\beta NL_w\ell_{g,1}^2 + 2\beta\ell_{f,0}\ell_{g,2} + L_F\ell_{g,1}^2\beta^2}{\mu_g}\right)\mathcal{R}_y(x^k, y^k)
\end{aligned}
$$

$$
\begin{aligned}
\overset{(b)}{=} & -\left(\frac{\alpha}{4} - \frac{L_w^2}{2\eta_2}\frac{\alpha^2}{\beta} - \frac{L_F}{2}\alpha^2 - \ell_{g,1}L_w\alpha^2 - \frac{c(\ell_{g,1}+L_g)}{2}\alpha^2\right)\|d_x^k\|^2 + \frac{\alpha\ell_{g,1}^2}{2}b_k^2 \\
& -\left(\frac{c\beta\mu_g}{2} - \frac{2c^2\alpha\ell_{g,1}^2 + \beta\eta_2\ell_{g,1}^2 + L_F\ell_{g,1}^2\beta^2}{\mu_g}\right)\mathcal{R}_y(x^k, y^k)
\end{aligned}
$$

$$
\begin{aligned}
\overset{(c)}{\leq} & -\left(\frac{\alpha}{4} - \frac{L_w^2\mu_g^2}{16\eta_2\ell_{g,1}^2}\alpha - \frac{L_F}{2}\alpha^2 - \ell_{g,1}L_w\alpha^2 - \frac{c(\ell_{g,1}+L_g)}{2}\alpha^2\right)\|d_x^k\|^2 + \frac{\alpha\ell_{g,1}^2}{2}b_k^2 \\
& -\left(\frac{c\beta\mu_g}{4} - \frac{\beta\eta_2\ell_{g,1}^2 + L_F\ell_{g,1}^2\beta^2}{\mu_g}\right)\mathcal{R}_y(x^k, y^k)
\end{aligned}
$$

$$
\begin{aligned}
\overset{(d)}{\leq} & -\left(\frac{\alpha}{8} - \frac{L_F}{2}\alpha^2 - \ell_{g,1}L_w\alpha^2 - \frac{c(\ell_{g,1}+L_g)}{2}\alpha^2\right)\|d_x^k\|^2 + \frac{\alpha\ell_{g,1}^2}{2}b_k^2 \\
& -\left(\frac{c\beta\mu_g}{8} - \frac{L_F\ell_{g,1}^2\beta^2}{\mu_g}\right)\mathcal{R}_y(x^k, y^k)
\end{aligned}
$$

$$
\overset{(e)}{\leq} -\frac{\alpha}{16}\|d_x^k\|^2 - \frac{c\beta\mu_g}{16}\mathcal{R}_y(x^k, y^k) + \frac{\alpha\ell_{g,1}^2}{2}b_k^2. \tag{42}
$$

where (a) comes from setting $\eta_1 = 2c\ell_{g,1}$, (b) is achieved by $c \geq \max\left\{\frac{8\ell_{g,2}\ell_{f,0}}{\mu_g^2}, \frac{8NL_w\ell_{g,1}^2}{\mu_g^2}\right\}$, (c) is gained by letting $\frac{c\alpha}{\beta} \leq \frac{\mu_g^2}{8\ell_{g,1}^2}$, (d) is achieved by

$$
\eta_2 = \frac{L_w^2\mu_g^2}{2c\ell_{g,1}}, \quad c \geq 2L_w\sqrt{\ell_{g,1}}
$$

and (e) is earned by

$$
\alpha \leq \frac{1}{8(L_F + 2\ell_{g,1}L_w + c(L_g + \ell_{g,1}))}, \qquad \beta \leq \frac{c\mu_g^2}{16L_F\ell_{g,1}^2}.
$$

In a word, a sufficient condition for (42) is

$$
c = \max\left\{\frac{8\ell_{g,2}\ell_{f,0}}{\mu_g^2}, \frac{8NL_w\ell_{g,1}^2}{\mu_g^2}, 2L_w\sqrt{\ell_{g,1}}\right\}, \qquad \eta_1 = 2c\ell_{g,1}, \qquad \eta_2 = \frac{L_w^2\mu_g^2}{2c\ell_{g,1}}
$$

$$
\beta \leq \min\left\{\frac{1}{\ell_{g,1}}, \frac{c\mu_g^2}{16L_F\ell_{g,1}^2}, \frac{c\ell_{g,1}^2}{\mu_g^2(L_F + 2\ell_{g,1}L_w + c(L_g + \ell_{g,1}))}\right\}, \quad \alpha \leq \frac{\mu_g^2}{8c\ell_{g,1}^2}\beta
$$

Then rearranging the terms and telescoping yield

$$
\frac{1}{K}\sum_{k=0}^{K-1}\|d_x^k\|^2 \leq \frac{8\ell_{g,1}^2}{K}\sum_{k=0}^{K-1}b_k^2 + \frac{16\mathbb{V}^0}{\alpha K} \overset{(a)}{\leq} 16(1 - \rho\sigma_g^2)^T\frac{\ell_{f,0}\ell_{g,1}^2}{\mu_g} + \frac{16\mathbb{V}^0}{\alpha K}
$$

$$\frac{1}{K}\sum_{k=0}^{K-1}\mathcal{R}_y(x^k,y^k) \le \frac{8\ell_{g,1}^2\alpha}{c\beta\mu_g K}\sum_{k=0}^{K-1}b_k^2 + \frac{16\mathbb{V}^0}{c\beta\mu_g K} \overset{(a)}{\le} 16(1-\rho\sigma_g^2)^T\frac{\alpha\ell_{f,0}\ell_{g,1}^2}{c\beta\mu_g^2} + \frac{16\mathbb{V}^0}{c\beta\mu_g K}$$

where (a) comes from Lemma 5. Then by choosing $T = \log(K)$, we know that

$$\frac{1}{K}\sum_{k=0}^{K-1}\|d_x^k\|^2 \le \mathcal{O}\left(\frac{1}{K}\right), \qquad \frac{1}{K}\sum_{k=0}^{K-1}\mathcal{R}_y(x^k,y^k) \le \mathcal{O}\left(\frac{1}{K}\right). \tag{43}$$

Moreover, since $\nabla_w\mathcal{L}(x^k,y^k,w)$ is $\ell_{g,1}^2$-Lipschitz smooth over $w$ according to Proposition 2 and

$$\nabla_w\mathcal{L}(x^k,y^k,w^\dagger(x^k,y^k)) = 0$$

then it holds that

$$
\begin{aligned}
\|\nabla_w\mathcal{L}(x^k,y^k,w^k)\|^2 &= \|\nabla_w\mathcal{L}(x^k,y^k,w^k) - \nabla_w\mathcal{L}(x^k,y^k,w^\dagger(x^k,y^k))\|^2 \\
&\le \ell_{g,1}^2\|w^k - w^\dagger(x^k,y^k)\|^2 \\
&\le 2\ell_{g,1}^2\|w^k - w^\dagger(x^{k-1},y^k)\|^2 + 2\ell_{g,1}^2\|w^\dagger(x^{k-1},y^k) - w^\dagger(x^k,y^k)\|^2 \\
&\le 2\ell_{g,1}^2 b_{k-1}^2 + 2\ell_{g,1}^2 L_w^2\alpha^2\|d_x^{k-1}\|^2.
\end{aligned}
$$

Therefore, averaging the left-hand-side and plugging in the gradient of $\mathcal{L}$ yield,

$$
\begin{aligned}
\frac{1}{K}\sum_{k=0}^{K-1}\mathcal{R}_w(x^k,y^k,w^k) &= \frac{1}{K}\sum_{k=0}^{K-1}\|\nabla_w\mathcal{L}(x^k,y^k,w^k)\|^2 \\
&\le \mathcal{O}\left(\frac{1}{K}\right) + \frac{2\ell_{g,1}^2 L_w^2\alpha^2}{K}\sum_{k=0}^{K-2}\|d_x^k\|^2 + \frac{1}{K}\|\nabla_w\mathcal{L}(x^0,y^0,w^0)\|^2 \\
&\le \mathcal{O}\left(\frac{1}{K}\right). 
\end{aligned} \tag{44}
$$

where the last inequality is due to (43) and the boundedness of $\|\nabla_w\mathcal{L}(x^0,y^0,w^0)\|^2$.

Similarly, we can bound

$$
\begin{aligned}
&\|\nabla_x f(x^k,y^k) + \nabla_{xy}^2 g(x^k,y^k)w^k\|^2 \\
&\le 2\|d_x^{k-1}\|^2 + 2\|\nabla_x f(x^k,y^k) + \nabla_{xy}^2 g(x^k,y^k)w^k - d_x^{k-1}\|^2 \\
&\le 2\|d_x^{k-1}\|^2 + 4\|\nabla_x f(x^k,y^k) - \nabla_x f(x^{k-1},y^k)\|^2 + 4\|\nabla_{xy}^2 g(x^k,y^k) - \nabla_{xy}^2 g(x^{k-1},y^k)\|^2\|w^k\|^2 \\
&\le 2\|d_x^{k-1}\|^2 + 4\ell_{f,1}^2\alpha^2\|d_x^{k-1}\|^2 + 4\ell_{g,2}^2\alpha^2\|d_x^{k-1}\|^2\|w^k\|^2 \\
&\overset{(a)}{\le} 2\|d_x^{k-1}\|^2 + 4\ell_{f,1}^2\alpha^2\|d_x^{k-1}\|^2 + 4\ell_{g,2}^2\alpha^2\|d_x^{k-1}\|^2\left(\frac{2\ell_{f,0}^2}{\sigma_g^2} + \mathcal{O}\left(\frac{1}{K}\right)\right) \\
&\le \left(2 + 4\ell_{f,1}^2\alpha^2 + 4\ell_{g,2}^2\alpha^2\left(\frac{2\ell_{f,0}^2}{\sigma_g^2} + \mathcal{O}\left(\frac{1}{K}\right)\right)\right)\|d_x^{k-1}\|^2
\end{aligned}
$$

where (a) comes from $\|w^k\|^2 \le 2\|w^\dagger(x^{k-1},y^k)\|^2 + 2b_{k-1}^2 \le \frac{2\ell_{f,0}^2}{\sigma_g^2} + \mathcal{O}(1/K)$. Therefore, using (43), we know

$$\frac{1}{K}\sum_{k=0}^{K-1}\mathcal{R}_x(x^k,y^k,w^k) = \frac{1}{K}\sum_{k=0}^{K-1}\|\nabla_x f(x^k,y^k) + \nabla_{xy}g(x^k,y^k)w^k\|^2 \le \mathcal{O}\left(\frac{1}{K}\right). \tag{45}$$

$\square$

# G   Additional Experiments and Details

**Synthetic experiments.** We test our algorithm GALET on Example 1 with different initialization. To see whether our stationary metric is a necessary condition of the optimality of BLO, we need to

define a value directly reflects whether the algorithm achieves the global optimality or not. By the explicit form of the solution set in (7), the global optimality gap can be measured by

$$\|x - 0.5\|^2 + \|0.5 + y_1 - \sin(y_2)\|^2 \tag{46}$$

where the first term correpsonds the distance to the global optimal $x^* = 0.5$ and the second term represents LL optimality. GALET achieves the global minimizers in (7) regardless of the initialization; see the top plot in Figure 1. The bottom plot in Figure 1 shows the stationary score of reformulation (2) and (3). If the KKT conditions of (2) hold, then it is equivalent to say the following

$$\nabla_x f(x, y) = 0, \quad \nabla_y f(x, y) = 0, \quad g(x, y) - g^*(x) = 0. \tag{47}$$

whose proof is attached below.

*Proof.* First, the KKT conditions of (2) can be written as there exists $\sigma^*$ such that

$$\nabla_x f(x^*, y^*) + \sigma^*(\nabla_x g(x^*, y^*) - \nabla g^*(x^*)) = 0 \tag{48a}$$
$$\nabla_y f(x^*, y^*) + \sigma^* \nabla_y g(x^*, y^*) = 0 \tag{48b}$$
$$g(x^*, y^*) - g^*(x^*) = 0 \tag{48c}$$

(48c) implies $y^* \in S(x^*)$. For PL function $g(x, y)$, we know that if $y^* \in S(x^*)$, then $\nabla_y g(x^*, y^*) = 0$ and according to Lemma 8, we have $\nabla_x g(x^*, y^*) - \nabla_x g^*(x^*) = 0$. Therefore, (48a) and (48b) are reduced to

$$\nabla_x f(x^*, y^*) = 0, \quad \nabla_y f(x^*, y^*) = 0$$

which completes the proof. □

Therefore, we can use the stationary score defined by $\|\nabla_x f(x, y)\|^2 + \|\nabla_x f(x, y)\|^2 + (g(x, y) - g^*(x))$ to measure whether the point achieves the KKT point of (2). For reformulation (3), our stationary measure is simply defined as the summation of the residual in (8). For both initializations, the stationary score of the value function-based reformulation (2) does not reach 0, whereas our measure based on the gradient-based reformulation (3) does. As the optimality gap of our algorithm converges to 0, this implies the KKT condition for the value function-based reformulation (2) does not serve as a necessary condition for optimality. In contrast, our stationary metric does, thereby validating our theoretical findings in Section 2.

We also test the sensitivity of the parameters of our method and show the trajectories starting from the same initialization in Figure 4. We conduct experiments on different parameter and the search grid is: UL stepsize $\alpha \in \{0.1, 0.3, 0.5, 0.7, 0.9\}$; shadow implicit gradient loops and stepsizes $T \in \{1, 5, 50\}, \gamma \in \{0.1, 0.5\}$ and LL stepsizes and loops $N \in \{1, 5\}, \beta \in \{0.1, 0.5, 1, 3\}$. The default choice of parameter is $\alpha = 0.3, K = 30, \beta = 1, N = 1, \gamma = 0.1, T = 1$. We observe that our algorithm is least sensitive to the shadow implicit gradient level parameters since all of the combinations of $T$ and $\gamma$ lead to almost the same trajectory. For the LL parameters $\beta$ and $N$, our algorithm is also robust as long as the learning rate $\beta$ is neither set too large ($\beta = 3$, divergence) nor too small ($\beta = 0.1$, slow convergence). The UL stepsize plays an important role on the convergence of GALET as different choices of $\alpha$ lead to distinct paths of GALET. The choices of $\alpha = 0.1, 0.3, 0.5$ all lead to the convergence of GALET, while a slightly larger value ($\alpha = 0.5$) causes GALET to prioritize optimizing the UL variable $x$ first.

**Real-data experiments.** We compare our method with the existing methods on the data hyper-cleaning task using the MNIST and the FashionMNIST dataset [21]. Data hyper-cleaning is to train a classifier in a corrupted setting where each label of training data is replaced by a random class number with a corruption rate $p_c$ that can generalize well to the unseen clean data. Let $x$ be a vector being trained to label the noisy data and $y$ be the model weight and bias, the objective function is given by

$$\min_{x, y \in S(x)} f(x, y^*(x)) \triangleq \frac{1}{|\mathcal{D}_{\text{val}}|} \sum_{(u_i, v_i) \in \mathcal{D}_{\text{val}}} \text{CE}(y^*(x); u_i, v_i)$$

$$\text{s.t. } y \in S(x) = \arg\min_y g(x, y) \triangleq \frac{1}{|\mathcal{D}_{\text{tr}}|} \sum_{(u_i, v_i) \in \mathcal{D}_{\text{tr}}} [\sigma(x)]_i \text{CE}(y; u_i, v_i)$$

where CE denotes the cross entropy loss and $\sigma$ denotes sigmoid function. We are given 5000 training data with corruption rate 0.5, 5000 clean validation data and 10000 clean testing data. The existing

methods for nonconvex-strongly-convex BLO [21, 32] often adopts a single fully-connected layer with a regularization as the LL problem, but the regularization and the simple network structure always degenerate the model performance. As our algorithm is able to tackle the nonconvex LL problem with multiple solutions, we can consider more complex neural network structure such as two layer MLP without any regularization. Since training samples is fewer than the model parameters, the LL training problem is an overparameterized neural network so that satisfies the PL condition.

**Algorithm implementation details.** Instead of calculating Hessian matrix explicitly which is time-consuming, we compute the Hessian-vector product via efficient method [53]. Specifically, we can auto differentiate $\nabla_y g(x, y)^\top w$ with respect to $x$ and $y$ to obtain $\nabla^2_{xy} g(x, y)w$ and $\nabla^2_{yy} g(x, y)w$. Also, let $v = \nabla_y f(x, y) + \nabla^2_{yy} g(x, y)w$ which detaches the dependency of $v$ over $x$ and $y$, the Hessian-vector product $\nabla^2_{yy} g(x, y)v$ can be calculated by auto differentiation of $\nabla_y g(x, y)^\top v$.

**Parameter choices.** The dimension of the hidden layer of MLP model is set as 50. We select the stepsize from $\alpha \in \{1, 10, 50, 100, 200, 500\}, \gamma \in \{0.1, 0.3, 0.5, 0.8\}$ and $\beta \in \{0.001, 0.005, 0.01, 0.05, 0.1\}$, while the number of loops is chosen from $T \in \{5, 10, 20, 30, 50\}$ and $N \in \{5, 10, 30, 50, 80\}$.

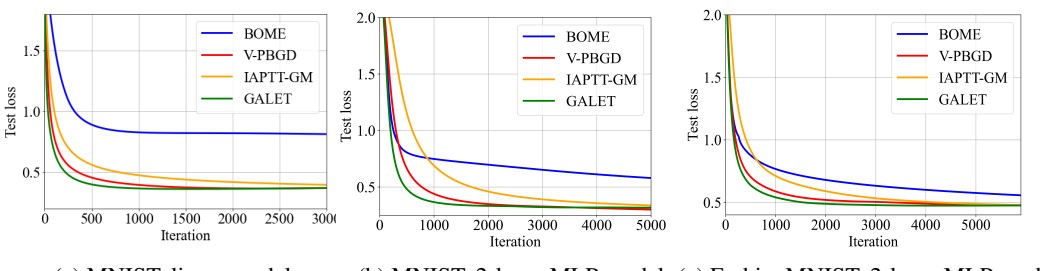

| (a) MNIST, linear model | (b) MNIST, 2-layer MLP model | (c) FashionMNIST, 2-layer MLP model |

Figure 7: Test loss v.s. iteration.

| Method | MNIST | | FashionMNIST |
|---|---|---|---|
| | Linear model | MLP model | MLP model |
| RHG [21] | $87.83 \pm 0.17$ | $87.95 \pm 0.21$ | $81.87 \pm 0.31$ |
| BOME | $88.76 \pm 0.15$ | $90.01 \pm 0.18$ | $83.81 \pm 0.26$ |
| IAPTT-GM | $90.13 \pm 0.13$ | $90.86 \pm 0.17$ | $83.50 \pm 0.28$ |
| V-PBGD | $90.24 \pm 0.13$ | $92.16 \pm 0.15$ | $84.18 \pm 0.22$ |
| **GALET** | $90.20 \pm 0.14$ | $91.48 \pm 0.16$ | $84.03 \pm 0.25$ |

Table 3: Comparison of the test accuracy of different methods on two dataset with different network structure. The results are averaged over 10 seeds and $\pm$ is followed by the variance.

The test loss v.s. iteration of different methods is shown in Figure 7, it can be seen that GALET converges fast in all of regimes. Moreover, we report the test accuracy of different methods in Table 3. It shows that GALET achieves comparable test accuracy with other nonconvex-nonconvex bilevel methods, which improves that of nonconvex-strongly-convex bilevel method RHG.

