# OpenReview forum: "An Alternating Optimization Method for Bilevel Problems under the Polyak-Łojasiewicz Condition"
_NeurIPS.cc/2023/Conference — NeurIPS 2023 poster_

### Official Review · Reviewer_k2BT · 2023-06-16

**Soundness:** 2 fair
**Presentation:** 2 fair
**Contribution:** 2 fair
**Rating:** 3
**Confidence:** 5

**Summary:**

This paper proposed a generalized alternating method for nonconvex bilevel optimization with PL condition in lower level. Meanwhile, it provided convergence analysis for the proposed method under a new stationary metric. Some experimental results demonstrate efficiency of the proposed method.

**Strengths:**

This paper proposed a generalized alternating method for nonconvex bilevel optimization with PL condition in lower level. Meanwhile, it provided convergence analysis for the proposed method under a new stationary metric. Some experimental results demonstrate efficiency of the proposed method.

**Weaknesses:**

In the paper, some incomprehensible words make it difficult for me to judge whether the conclusion in the paper is correct ?

For example, clearly, the lines 276-277 of the paper (‘’ Note that $\lambda_g>0$ allows $\nabla^2_{yy}g(x,y)$ to have either negative eigenvalues or zero eigenvalues … .’’) are not incorrect.

From Table 1, the authors said BOME [33] and MGBiO [24] only studied a singleton on $S(x)$. From [33] and [24], they studied the bilevel optimization problem:

$$  \min_{x\in \mathbb{R}^{d_x}} f(x,y^*), \quad s.t., \quad y^* \in \arg\min_{y\in \mathbb{R}^{d_y}} g(x,y),$$

which equals the bilevel problem (1) given in this paper. Thus, I think the BOME [33] and MGBiO [24] also studied non-singleton on $S(x)$.

Form this paper, the authors point that they introduce and use a new stationary metric in convergence analysis. It is better? Please detail differences between it and other metrics used in [48,33,24,2,38].

**Questions:**

1)	In Assumption 2, since $\lambda_g>0$, clearly, $\nabla^2_{yy} g(x,y)$ only have positive eigenvalues ?

2)	In the equality (11b), can we use
$$\nabla^2_{yy} g(x^k,y^{k+1})\big(\nabla_y f(x^k,y^{k+1})+\nabla^2_{yy} g(x^k,y^{k+1})w^{k,t}\big)$$ instead of
$$\nabla_y f(x^k,y^{k+1})+\nabla^2_{yy} g(x^k,y^{k+1})w^{k,t}$$ ?

3)	In the proposed GALET algorithm, since using the multi-step gradient descent to update variable $y$ and introduced variable $w$, why still compute Hessian matrix $\nabla^2_{yy} g(x,y)$ and Jacobian matrix $\nabla^2_{xy} g(x,y)$ ?

4)	In Theorem 2, how to choose the learning rates $\alpha$, $\rho$ and $\beta$ ? Please detail them.

5)	In Theorem 2, why give $S=O(1)$ and $T=O(\log(K))$ ? In other words, why $T$ is larger than $S$ ?



---------------------------------------After Rebuttals ------------------------------------------------------------------


Thanks for authors' responses. However, the authors' responses did not address my main concern: Assumption 2 is very strict.

Adding Assumption 1, i.e. $ \nabla^2_g$ is Lipschitz continuity, the authors claim that Hessian matrix maybe includes zero eigenvalue (singular value). Clearly, $\lambda_g$ is very small even approach zero. Thus, the gradient complexity is infinite, **Please the table given in Rebuttal by Authors to Reviewer swWK**.

When Hessian matrix does not include zero eigenvalue (singular value), i.e. $\lambda_g >0$,
clearly, assumptions 2  and 1 are exactly strong convexity, defeating the whole purpose of the paper.

Thus I lower my score.

**Limitations:**

Not potential negative societal impact.

---

> ### Author Rebuttal · Authors · 2023-08-08
>
> We thank the reviewer for the careful reviews. However, we feel the reviewer might have misunderstood some technical points in our paper which leads to the concern on the correctness of the conclusion in the paper. Our response to your comments follows.
>
> **Q1: Does $\lambda_g>0$ implies positive eigenvalue?**
>
>  We answer it in the **General Response G2**. Hope this can address your concern.
>
> **Q2: Singleton assumptions in BOME [33] and MGBiO [24]**
>
> Thank you for pointing out this. We are very sorry for the inaccurate description of the related work. Upon double checking BOME [33], we found that it considers two settings: for the singleton scenario, it has a faster rate of ${\cal O}(\epsilon^{-3})$; and for the nonsingleton setting, it has a slower rate of ${\cal O}(\epsilon^{-4})$; both of which are slower than our rate. We will add this note in our submission. For MGBiO [24], they only consider the singleton case, otherwise $y^*(x)=\arg\min_y g(x,y)$, then $f(x,y^*(x))$ would be multivalued, rendering $\min_x f(x,y^*(x))$ ill-posed and the implicit function theorem would fail. Actually, we adopt the optimistic bilevel optimization formulation in [55,37,2,43]
> $$\min_{x,y\in S(x)}f(x,y),\qquad \textrm{s.t. } S(x)=\arg\min_y g(x,y)$$
> instead of the singleton bilevel formulation.
>
>
> **Q3: Whether the new stationary metric is better.**
>
> 1. The metric in [33,24,2] cannot be applied here.  As we stated in Line 51-55, the proposed stationary metric is a **generalized version** of the existing measure in [33,24,2]. Due to the space limitation, we have mentioned it in Line 188-190 but deferred the detailed discussions in Appendix H.1.
> 2. [48,38] relax the lower-level problem by its $\delta$- approximate solution or pessimistic trajectory with initialization auxiliary, so the stationary point they achieved are the $\epsilon$ stationary point of the $\delta$- approximate problem, i.e. $(\epsilon,\delta)$- stationary point of the original BLO. Instead, we proved that (8b) are the necessary conditions to the optimality of the original BLO problem, so the stationary point we achieved is the $\epsilon$ stationary point to the original BLO problem.
>
> We also highlight the importance of generalizing the existing measure without additional CQs in Section 2.3.
>
> **Q4: Removing the additional Hessian in the update (11b).**
>
> Thank you for raising this question. We answer it in the **General Response G1**.
>
> **Q5: The reason for computing Hessian and Jacobian.**
>
> Implementing GALET does not require explicitly computing Hessian and Jacobian because we calculate Hessian-vector and Jacobian-vector product instead as shown in Line 760-764 in Appendix and they can be easily computed by auto differential function in pytorch or backpropagation. Also computing Hessian-vector and Jacobian-vector product are standard even in bilevel with strongly convex LL problems such as in [8, 9, 23, 20, 28, 31, 30, 1, 11].
>
> **Q6: Choices of learning rates $\alpha,\beta,\rho$.**
>
> We answer it in the **General Response G3**. Hope this can address your concern.
>
> **Q7: Reasons for $T$ larger than $S$.**
>
> In Theorem 2 of our paper, it appears there is no $S$. We guess that you meant to refer to $N$ instead. We set $N={\cal O}(1)$ because we use warm start strategy on the update of $y$, i.e. update $y^{k,n}$ starting from $y^k$, but use cold start on the update of $w^{k,t}$ initialized at $0$. To guarantee the convergence of $\\{w^{k,t}\\}\_{t=0}^\infty$ to the minimum-norm solution, one need $T={\cal O}(\log(K))$ iterations. The warm strategy is not suitable for $w$-update in PL setting, which is actually one of the key technical challenges we overcome, due to the following reasons.
>
> 1. **(Bounded trajectory)** Different from lower-level objective $g(x,y)$, the carefully designed objective $\mathcal{L}(x,y,w)$ in shadow implicit gradient level is only Lipschitz smooth over $x$ and $y$ for bounded $w$. Therefore, one should ensure the boundedness of $w^{k,t}$ sequence. We achieve this by initializing $w^{k,t}$ from $0$ (cold start strategy) such that it converges to the minimum-norm solution which is bounded.
> 2. **(Lipschitz continuous trajectory)** Characterizing the upper-level descent requires the Lipschitz continuity of the limit point of the trajectory of $w^{k,t}$ over $x$ and $y$ (i.e. the Lipschitz continuity of $w^\dagger(x,y)$ over $x$ and $y$). But the warm start strategy yields a trajectory of $w$ that is not Lipschitz continuous over $x$ and $y$.
>
>
> With the above clarifications, we sincerely hope the reviewer can reconsider the correctness of our theoretical results!  We are happy to answer any follow-up questions on the correctness.

---

### Official Review · Reviewer_wxYg · 2023-07-06

**Soundness:** 4 excellent
**Presentation:** 3 good
**Contribution:** 3 good
**Rating:** 7
**Confidence:** 3

**Summary:**

This work studies an important problem of bilevel optimization with non-convex lower level function. The authors reformulate the problem as a constraint optimization and prove a constraint qualification (CQ) for the reformulated problem under mild assumptions, and show that other stronger CQ do not hold under the same assumptions. Next, authors introduce the stationarity measure for this problem and propose an algorithm which converges in this measure with minimax optimal rate under the given assumptions. Numerical simulations on synthetic and real data are provided to test the proposed algorithm and compare to other baselines.

**Strengths:**

The paper is well written and easy to follow. The limitations of previous works are well explained and the goal of the paper is clearly stated. The key contribution of the paper is substantially relaxing the assumptions and proving improved iteration complexity for bilevel problems, which matches with the optimal complexity (even for single level problems) in terms of $\epsilon$.

**Weaknesses:**

1. The mathematical clarity of some statements in the main part can be certainly improved. For instance, in Definition 2 (calmness) it is unclear to me if h(x, y) is a function or a map to $\mathbb R^d$, is $q$ a vector or a scalar? I also assume that the statement should hold for any pair (x', y'), not only for one. Although one can eventually infer the answers to these questions from the proofs in the appendix, it is better to be more rigorous.

2. In Figure 2, I do not understand what is the difference between the blue and the red boxes (Gradient-based PL-BLO reformulation).

3. I suggest to remove the right plot from Figure 5. It does not really make sense to compare the actual convergence rate with the asymptotic theoretical sublunar rate. How the exact constant was chosen? Why GALET is actually faster than theoretically proven 1/K.

4. The intuition for convergence of terms $\mathcal R_x$ and $\mathcal R_y$ are well explained before stating Theorem 2. However, why $\mathcal R_w$ converges is not immediately unclear from (20). I suggest to clarify this in the revision.

**Questions:**

1. In the proof of Lemma 3, do we need smoothness of $f(x, \cdot)$? It does not seem to be used in the proof.

2. In theorem 1 (8b), it is unclear to me where the term $\nabla_{yy}^2 g(x^*, y^*)$ comes from. Is it necessarily positive definite? It would be good to explain why this term is needed. Same question for $\nabla_{yy}^2 g(x^k, y^{k+1})$ term in (11b). The authors claim "the additional Hessian in (11b) is inevitable". Why?

3. Why do we necessarily have $\lambda_g >0$? for any $x, y$ the Hessian might have negative eigenvalues for PL functions.

**Limitations:**

The paper makes the fist steps in relaxing assumptions and proving the optimal rate for the fundamental setting of bilevel nonconvex optimization. Authors admit that the extension of the developed techniques to the stochastic setting might require different techniques.

---
Acknowledgement of the authors' rebuttal.

I thank the authors for the rebuttal and addressing my concerns. It is a little disappointing that in the discussion phase it turned out that the lower level problem cannot be non-convex PL while it was one of the key claims of the paper (it's of course great that other reviewers could detect this during the review period). I believe the key results of this work are still valuable even if they merely apply for convex lower level problems and the condition on the eigenvalues does not seem very natural. I still recommend the acceptance of this work, but am less certain about the decision after the rebuttal/discussion.

---

> ### Author Rebuttal · Authors · 2023-08-08
>
> We thank the reviewer for appreciating our contribution and the constructive review. Our response to your comments follows.
>
>
> **Q1: Rigorous definition of calmness.**
>
> Thank you for your suggestions and we will formalize the definition of calmness. To answer your specific confusion, $h(x,y)$ can be function or map and the dimension of $q$ is the same as the output of $h(x,y)$. If $h(x,y)=g(x,y)-g^*(x)$, which is a function, then $q$ is a scalar; if $h(x,y)=\nabla_yg(x,y)$, which is a map, than $q$ is a vector. The statement should hold for any $(x^\prime, y^\prime)$.
>
> **Q2: Typo in Figure 2.**
>
> Sorry for the confusion. We made a typo in Figure 2 and the term in red boxes should be "Gradient based strongly convex BLO reformulation". We will correct it.
>
> **Q3: Removing the right plot in Figure 5.**
>
> The purpose of Figure 5 is to show the convergence rate of GALET, with respect to the stationary measure, is approximately ${\cal O}(1/K)$, as the test loss curve does not directly represent the stationary measure.  We choose the constant of ${\cal O}(1/K)$ to make it close to the curve of GALET for comparison. We can remove it if it is not suitable.
>
> **Q4: Explain the reason why $\mathcal{R}_w$ converges.**
>
> The reasons are that $\lim_{t\rightarrow\infty}w^{k,t}=w^\dagger(x^k,y^{k+1})$ and $\mathcal{R}_w(x^k,y^{k+1},w^\dagger(x^k,y^{k+1}))=0$ according to the stationary condition of $\mathcal{L}(x,y,w)$ with respect to variable $w$.
>
> **Q5: Smoothness of $f(x,\cdot)$ in Lemma 3.**
>
> In the proof of Lemma 3, it does not rely on the smoothness of $f(x,\cdot)$ but the smoothness of $g(x,\cdot)$. The smoothness together with PL condition of $g(x,\cdot)$ ensures the error bound condition, which plays an important role in the proof of Lemma 3.
>
> **Q5: Additional Hessian in (8b) and (11b).**
>
> Thank you for raising this question. We answer it in the **General Response G1**.
>
>
> **Q6: Does $\lambda_g>0$ implies nonnegative eigenvalue?**
>
> Thank you for raising this clarification question. Since it is also raised by other reviewers, we answer it in the **General Response G2**.
>
> Thanks for the careful review and giving favorable recommendation.

---

### Official Review · Reviewer_5R75 · 2023-07-07

**Soundness:** 3 good
**Presentation:** 3 good
**Contribution:** 2 fair
**Rating:** 5
**Confidence:** 5

**Summary:**

The paper studies unconstrained bilevel optimization under the PL condition at the lower level. All previous work is limited to strongly-convex lower-level problems. The authors propose a new convergence criterion for this setting based on stationary-point seeking reformulation, supporting it with the calmness and constraint-qualification arguments. The algorithm is a slight variant of recent Hessian-inverse-free methods for strongly-convex lower-level cases. Convergence analysis is performed with standard smoothness assumptions in deterministic settings (with access to true gradients and Hessians). The algorithm is demonstrated via the MNIST data-hyperclearning experiment.

**Strengths:**

The paper extends the existing literature to problems with PL lower level. Bilevel optimization is a fast-moving area, and this extension is a timely contribution to the literature. I haven't seen yet the PL condition considered in the context of Bilevel optimization (even though it appeared several times in min-max literature, and the underlying idea is not quite different).

- The convergence criterion is well-supported by calmness and constraint-qualification arguments.

- Presentation is clear and easy to follow.

**Weaknesses:**

Overall, the algorithmic idea is essentially the same as previously developed algorithms.

- For instance, the work in [11] considered the same algorithm with strongly-convex lower-level objectives, to avoid Hessian-inverse estimation. The idea is essentially to solve a linear equation efficiently, which can be done in various ways (e.g., by running the GD on a quadratic function). This work does the same, the only difference is to force the initializer $w_0$ to be at 0, so that GD converges to the minimum-norm solution (the pseudo-inverse solution). However, the same argument does not seem to be applicable to SGD or non-zero initialization. Furthermore, this seems why Assumption 2 is required: lower-bound for non-zero singular values for all $y$. Given that the lower-level objective is already $\mu$-PL, another assumption seems redundant.

- The paper only considers deterministic cases, but most previous work can also handle stochastic gradients and Hessian oracles. Though this paper seems to be hard to be extended to stochastic analysis.

- I think there are several technical issues that need more justification. Please see the questions below.

**Questions:**

- Can a PL function have multiple optimal solutions, and at the same time Hessian-Lipschitz? Because at suboptimal points that is a neighborhood of optimal points, the gradient-growth has to be at least $\mu$.

Otherwise, which one should be given up? Analysis might be intact if we change the condition to have a unique lower-level solution, but if we give up Hessian-Lipschitzness, I am a little not sure if everything still works.


- In the inner loop, w_0 already has to start from 0 to ensure the minimum-norm solution. What can be said if the gradient is not exact or if $w_0$ starts from other points?


- Can you at least have Assumption 2 only at the optimal points, and not on all y?

- For (8b) in KKT condition, why not just $||\nabla f(x^*, y^*) + \nabla^2_{yy} g(x^*, y^*) w^*||^2 \le \epsilon$, instead of having additional $\nabla^2_{yy} g(x^*,y^*)$?

**Limitations:**

This paper is theoretical and has no concerns about societal impact.

---

> ### Author Rebuttal · Authors · 2023-08-08
>
> We thank the reviewer for appreciating our presentation and recognizing our work as "a timely contribution to the literature." Our response to your comments follows.
>
> **Q1: Concerns on the algorithmic idea.**
>
> Thank you for recognizing the necessity to extend bilevel optimization (BLO) to the PL setting, but we want to highlight that it is very different from PL min-max problem as there are fundamental challenges in PL BLO.
>
> 1. **PL min-max problem does not have issue of invalid constraint qualification (CQ).** In sequential min-max problem where $g=-f$, once we have the gradient of value function $g^*(x)=\min_y g(x,y)$, a nature stationary condition is $\nabla g^*(x)=0$. This is quite different in BLO since the stationary criterion is not naturally defined and we need to find a provable CQ condition ensuring the (weak) KKT conditions are met at the optimality. Actually, sequential min-max problem is a special case of BLO where CQ automatically holds [60]. [33] also mentioned the importance of CQ to ensure the stationarity of BLO but they did not verify the CQ they assumed. We highlight the importance of the provable CQ condition in Section 2.3.
> 2. **New proof techniques.**
> **(a)** In min-max problem, previous studies characterize the drifting of the lower-level (LL) optimality residual $\mathcal{R}_y\left(x^{k+1}, y^k\right)-\mathcal{R}_y\left(x^k, y^k\right)$ in (16b) by the identical stationary conditions of $f$ and $g$ coming from $f=-g$. But in BLO, the stationary condition of $f$ and $g$ can be totally different. Instead, we derive (16b) by the smoothness of $g^*(x)$ and use the opposite sign of $g(x, y)$ and $g^*(x)$ ($g(x,y)-g^*(x)$) to cancel out the first order terms in Taylor expansion.
> **(b)** In min-max problem, a natural Lyapunov function for upper-level descent is $\max_y f(x,y)$, which is not the target objective in BLO. We adopt $f(x, y)+w^{\dagger}(x,y) \nabla_y g(x, y)$, which is originated from the KKT increment (8a). Moreover, our proofs are based on the multi-sequence update of the KKT increments instead of the identical stationary conditions of $f$ and $g$ in min-max problem.
>
> **Q2：Can a PL function have multiple optimal solutions and also Hessian-Lipschitz?**
>
> These assumptions are not contradictory. For example, given an UL variable $x$, a strongly convex and $\ell_h$ Hessian-lipschitz function $h(x,\cdot)$ and a matrix $A$ that is not in full rank, the function $g(x,y)=h(x,Ay)$ satisfies these three conditions. We justify it as follows:
>
> **1) PL over $y$.** This is proved by [29, Page 14 Appendix B].
>
> **2) Hessian Lipschitz.** The Hessian of $g(x,y)$ can be calculated by $\nabla_{yy}^2g(x,y)=A^\top\nabla^2h(x,Ay)A$, where $\nabla^2h(x,Ay)$ denotes for $\nabla^2h(x,z)|\_{z=Ay}$. Then the Hessian Lipschitz continuity of $g(x,\cdot)$ can be obtained from \begin{align*}
> \\|\nabla_{yy}^2g(x,y_1)-\nabla_{yy}^2g(x,y_2)\\|&\leq \\|A^\top(\nabla^2h(x,Ay_1)-\nabla^2h(x,Ay_2))A\\|\\\\
> &\leq \\|A\\|^3\ell_h\\|y_1-y_2\\|.
> \end{align*}
>
> **3) Multiple solutions.** $g(x,y)$ only depends on the value of $Ay$, so the null space component does not affect the value of $g$. If $A$ is not full rank, for any solution $y^*$ of $g$, $y^*+\operatorname{Ker}(A)$ is the solution space.
>
> **Q3：Can $w_0$ be initialized from other points or the gradient is not exact?**
>
> Thank you for this interesting question which may help us extend our analysis.
>
> 1. **$w^{0}$ can be other points other than $0$ if $w^{k,0}=w^0$ for any $k$.** We answer it in the **General Response G4**.
>
> 2. **The gradient can be inexact if the bias is bounded.** The convergence relies on the boundedness and Lipschitz continuity of limit points. Similar to the non-zero initialization, if the deterministic gradient is inexact but it has bounded bias, then the sequence $\\{w^{k,t}\\}\_{t=0}^\infty$ will converge to a bounded neighborhood of the exact solution. Lipschitz continuity of the biased solution can be earned similarly as the non-zero initialization.
>
> **Q4：Relax Assumption 2 to the optimal points**
>
> It can be very hard. Assumption 2 is a generalization of the strongly convex assumption in BLO in [8, 23, 20, 28, 1, 11] that has been assumed for any $y$. As it is used in per-iteration analysis of the algorithm, relaxing it may need major change of the analysis.
>
> **Q5：The use of additional Hessian in (8b).**
>
> Thank you for raising this question. We answer it in the **General Response G1**.
>
> **Q6: Possible extension to stochastic setting?**
>
> For BLO with a strongly convex LL problem, the stochastic setting has been actively studied. But most literature on BLO with nonconvex LL problems [37, 2, 43] focuses on the deterministic case. And even for deterministic case, the iteration complexity does not match that of single-level optimization.
>
> Therefore, in this work, we take the first step to address the fundamental issue if we can achieve the optimal convergence rate for the deterministic nonconvex BLO. Extending this work to a stochastic setting is of high interest yet also nontrivial that involves new analysis of bounding the higher-order stochastic moments (beyond the third order). We will develop new tools to address in our future work.
>
> With the above clarifications, we sincerely hope the reviewer can reconsider the theoretical contribution of our work and raise the score. Thank you again!

---

> > ### Comment · Reviewer_5R75 · 2023-08-21
> > **Rebuttal Response**
> >
> > I thank the authors for their detailed clarification. My concerns are mostly addressed. I raise my score to 5.

---

> > > ### Author Response · Authors · 2023-08-21
> > >
> > > It's great to know that your concerns are mostly solved and thank you for raising the score! We appreciate it.

---

### Official Review · Reviewer_swWK · 2023-07-18

**Soundness:** 3 good
**Presentation:** 3 good
**Contribution:** 3 good
**Rating:** 5
**Confidence:** 4

**Summary:**

This paper addresses the bilevel optimization problem, which may feature a non-convex lower-level problem with non-unique optimal function values. The lower-level problem can be transformed into a constraint optimization problem using function value or gradient-based constraints. The paper establishes necessary conditions via calmness to evaluate the optimality of bilevel optimization problems where the inner problem satisfies the PL condition. Based on these conditions, the authors develop the GALET method, which alternately updates the inner and outer variables. The effectiveness of GALET is demonstrated through an experiment involving hyper-data cleaning, which shows that GALET performs comparably to V-PBGD.

**Strengths:**

The main strength of this paper is the proposal of a stationary metric to evaluate the optimality of BLO problems with non-convex lower-level problems. This metric enables to achieve optimal iteration complexity.

**Weaknesses:**

Overall, I have the following concerns and comments about this paper:
1.  While formulation (2) is directly equivalent to BLO problem (1), the equivalence of the gradient-based formulation (3) only holds under PL conditions. As a result, it is unclear whether the proposed stationary condition is applicable in cases where the inner function of BLO does not satisfy the PL condition. Do you think it is feasible to extend the proposed stationary condition to the more general non-convex setting?

2. As checked, it appears that the values of $\beta$, $N$ $\alpha$, and $\eta_1$  are closely related in the analysis. However, in Theorem 2 and Lemma 4,  the authors merely simplify these parameters to the order of $\mathcal{O}(1)$. As a result, the theoretical results do not elucidate the relationship between these parameters. It would be helpful if the authors could provide more details on how these parameters are related to each other in the main results.

3. Figure 1 shows the stationary score of function value-based and gradient-based formulations (2) and (3) under different initializations. It is unclear which algorithm is adopted to solve the function value-based problems. It may affect the value of the KKT measure over iteration. Could you provide more information regarding the details of this result?

4.  The main results presented by the authors focus solely on iteration complexity and do not provide information on the gradient oracle complexity or Hessian-vector complexity. Could you provide the complexity of the gradient and Hessian vector? It would be beneficial to include information on how these complexities depend on various Lipschitz constants and the PL constant $\mu_g$. In addition, the strongly convex function is a special example of the PL function. It would be helpful to compare the results of GALET for strongly convex inner functions to those in the existing literature.

5. The proposed method, GALET, is a second-order method, while the V-PBGD method is a fully first-order method. The authors have claimed complexity improvements over V-PBGD. But, from the numerical experiments, I did not observe any advantages over the first-order V-PBGD method, as expected.  It raised the question of whether the second-order information helped in your experiments. It would be helpful if you could provide additional evidence to clarify the performance of the experiments.

**Questions:**

As mentioned in weaknesses.

**Limitations:**

Yes.

---

> ### Author Rebuttal · Authors · 2023-08-08
>
> We thank the reviewer for appreciating our contribution and the constructive review. Our response to your comments follows.
>
> **Q1: Possibility of extending the proposed stationary condition to the general non-PL nonconvex setting.**
>
> Thanks for raising this intriguing point! Currently, generalizing the proposed stationary condition to the general non-PL nonconvex setting is challenging. The difficulties primarily arise from three aspects: 1) The lower-level stationary points are not necessarily feasible for the BLO because they are not necessarily the lower-level optimal points; 2) identifying a valid CQ condition for the non-PL BLO becomes unknown; and 3) The PL condition is currently the most reliable condition, ensuring the existence of an algorithm - gradient descent - that converges to the global optimal sets at a linear rate. However, for a general nonconvex lower-level problem, finding the globally optimal lower-level solution set efficiently remains difficult, letting alone defining a stationary measure.
>
> **Q2: Explicitly show the choices of $\alpha,\beta,N,\eta_1$.**
>
> We answer it in the **General Response G3**. Hope this can address your concern.
>
> **Q3: Provide more information regarding Figure 1.**
>
> *Setting of Figure 1.* To generate Figure 1, we run GALET on the Example 1 and report two types of stationary score -- value function based and gradient based -- as a function of iteration index. Due to the space limitation, the calculation of two types of KKT score is detailed in the line 713-734 in Appendix.
>
> *Interpretation of Figure 1.* As the optimality gap (the distance to the global optimal set) of GALET converges to $0$, it means GALET achieves global optimal solutions of Example 1 under both initializations. However, the value function based stationary score does not reach 0, but the gradient-based stationary score does. This implies the KKT condition for the value function-based reformulation does not serve as a necessary condition for optimality.
>
> Thank you for raising this point. We will move the details from the Appendix directly into the main paper and add explanations.
>
> **Q4：Complexity of the gradient and Hessian vector and the dependence on Lipschitz constant and $\mu_g$.**
>
> Thank you for this constructive suggestion.
>
> Let $\operatorname{Gc}(f, \epsilon)$ and $\operatorname{Gc}(g, \epsilon)$ : number of gradient evaluations w.r.t. $f$ and $g$;  $\mathrm{JV}(g, \epsilon)$ : number of Jacobian-vector products $\nabla_{xy} g(x, y) v$; $\mathrm{HV}(g, \epsilon)$ : number of Hessian-vector products $\nabla_{yy}^2 g(x, y) v$. If we define the condition number as $\kappa_1=\ell_{g,1}/\mu_{g}$ and $\kappa_2=\ell_{g,1}/\lambda_{g}$, the gradient and Hessian-vector product complexity is summarized in Table R1.
> |Algorithm |$\operatorname{Gc}(f, \epsilon)$ | $\operatorname{Gc}(g, \epsilon)$ |$\operatorname{Jv}(g, \epsilon)$|$\operatorname{Hv}(g, \epsilon)$|
> | :---------- | :--------: | :-------: |:--------: | :-------: |
> |GALET  | ${\cal O}\left(\frac{\kappa_1^4\kappa_2^2 }{\min\\{\kappa_1,\kappa_2\\}}\epsilon^{-1}\right)$| ${\cal O}\left(\frac{\kappa_1^4\kappa_2^2 }{\min\\{\kappa_1,\kappa_2\\}}\epsilon^{-1}\right)$ | ${\cal O}\left(\frac{\kappa_1^4\kappa_2^2 }{\min\\{\kappa_1,\kappa_2\\}}\epsilon^{-1}\right)$| $\tilde{\cal O}\left(\frac{\kappa_1^4\kappa_2^2 }{\min\\{\kappa_1,\kappa_2\\}}\epsilon^{-1}\right)$ |
> |Strongly-convex GALET |${\cal O}\left(\kappa_1^5\epsilon^{-1}\right)$| ${\cal O}\left(\kappa_1^5\epsilon^{-1}\right)$|${\cal O}\left(\kappa_1^5\epsilon^{-1}\right)$|$\tilde{\cal O}\left(\kappa_1^5\epsilon^{-1}\right)$|
> |stocBiO [28]  |${\cal O}\left(\kappa_1^3\epsilon^{-1}\right)$| ${\cal O}\left(\kappa_1^4\epsilon^{-1}\right)$|${\cal O}\left(\kappa_1^3\epsilon^{-1}\right)$|${\cal O}\left(\kappa_1^{3.5}\epsilon^{-1}\right)$|
>
>
> *Table R1: Comparison of bilevel deterministic optimization algorithms.*
>
> The complexity of GALET in strongly-convex case is worse than that of deterministic strongly-convex case in stocBiO [28] in terms of $\kappa_1$, because GALET necessitates an additional Hessian in the $w$ update, which subsequently enlarges the Lipschitz continuity modulus of the objective in the shadow implicit gradient level. However, as discussed in Section 3.2, the additional Hessian is inevitable for cases that are not strongly convex.
>
> **Q5：Empirical performance versus V-PBGD**
>
> Great observation! Thank you for this insightful question. As shown in the left figure in Figure 5 and Figure 7 in the Appendix, GALET converges faster than V-PBGD in terms of iterations, thus verifying that GALET's convergence rate of ${\mathcal O}(\epsilon^{-1})$ surpasses that of V-PBGD, which is ${\mathcal O}(\epsilon^{-1.5})$. However, when it comes to test accuracy, GALET does not outperform V-PBGD due to the following factors that may influence test accuracy. We propose two possible explanations:
>
> 1. V-PBGD relaxes the lower-level exact solution by its $\epsilon$ approximate solution. This relaxation introduces errors into the original bilevel problem (from optimization perspective), but it actually helps prevent the overfitting issues in the simulated hyper-cleaning task as the lower-level problem is a model training problem.
> 2. The global optimal solutions of bilevel problem are correlated with validation accuracy (bilevel objective), rather than the test accuracy. Also, attaining the stationary points of the bilevel problem does not necessarily guarantee reaching the global optimum of the bilevel problem, letting alone with the test performance.
>
> This is also the reason that we plot the stationary score we proposed of GALET and show the iteration complexity of GALET nearly matches ${\mathcal O}(\epsilon^{-1})$ in the right figure in Figure 5. This result is more aligned with our improved theory over V-PBGD.
>
> We hope our clarification addresses your concerns. Thank you once again for your careful review!

---

> > ### Comment · Reviewer_swWK · 2023-08-21
> > **Response to Authors**
> >
> > The authors has almost addressed my concerns. Thank you.
> >
> > But the stationary condition is only applied to the problems that satisfies PL condition. In this sense, this work  is not exciting or interesting as I have expected.
> >
> > I understand the reasons that the test accuracy of GALET may not better than V-PBGD. I appreciate that the authors have proved the faster convergence rate. But in the machine learning community, we still expect their practical performance in the test accuracy. The  proposed algorithms can not make improvements over this measure target.

---

### Author Rebuttal · Authors · 2023-08-08

## General Response ##

We sincerely thank the reviewers for their constructive comments. All reviewers have agreed that our work has substantially relaxed the strongly convex lower-level assumptions in bilevel optimization and proved the optimal complexity in terms of $\epsilon$. Comments from all the reviewers were really helpful, which we believe have been fully addressed in detail in our rebuttal.

We first clarify the common issues raised by more than one reviewer.

**G1: The use of additional Hessian in (8b) and (11b).**

Thanks for raising this good question! Adding additional Hessian in the stationary metric and the update rule is actually the key of our algorithm design that enables GALET to converge for bilevel problem (BLP) with PL lower-level problems!

First, the additional Hessian appears in (11b) because it is the gradient of $\mathcal{L}(x^k,y^{k+1},w)$. If we were to redefine $\mathcal{L}(x,y,w)$ in a different manner, say $w^\top\nabla_y f(x, y)+w^\top\nabla_{y y}^2 g(x, y) w$, this newly defined objective would not satisfy PL condition for non-positive definite $\nabla_{yy}g(x,y)$. Consequently, no existing algorithm could guarantee a global linear convergence rate.

Second, as updating $w$ by the gradient of $\mathcal{L}(x,y,w)$ cannot guarantee that the null space component of $\nabla f(x^*, y^*)+\nabla_{y y}^2 g(x^*, y^*) w^*$, we project it to the kernel of Hessian of g, i.e., $\operatorname{Proj}\_{\operatorname{Ker}(\nabla_{yy}^2 g(x^*, y^*))}(\nabla_y f(x^*, y^*)+\nabla_{y y}^2 g(x^*, y^*) w^*)$ diminishes. Therefore, we adopt $\\|\nabla_{y y}^2 g(x^*, y^*)(\nabla_y f(x^*, y^*)+\nabla_{y y}^2 g(x^*, y^*) w^*)\\|\leq\epsilon$ in (11b) instead of $\\|\nabla f(x^*, y^*)+\nabla_{y y}^2 g(x^*, y^*) w^*\\|\leq\epsilon$. We prove it is still a necessary and tight condition for the stationary of BLO in Theorem 2.

Finally, if $\nabla_{yy}g(x^*,y^*)$ is invertible (e.g., in the strongly convex lower-level setting), with and without additional Hessian are equivalent.

**G2: Does $\lambda_g>0$ implies positive eigenvalue?**

No, it does not. The value of $\lambda_g$ is defined as as the lower bound of the minimal nonzero singular value of the Hessian matrix. Specifically, $\lambda_g=\inf_{x,y}\\{\lambda_{\min}^{\neq 0}(\nabla_{yy}^2g(x,y))\\}$, where $\lambda_{\min}^{\neq 0}(\nabla_{yy}^2g(x,y))$ refers to the smallest nonzero singular value of the Hessian.

**It's important to differentiate the singular value from the definition of an eigenvalue.** For any matrix $A\in\mathbb{R}^{d\times d}$, singular values of $A$ are the square roots of the eigenvalue of $A^\top A$ so that they are always nonnegative. This is different from eigenvalues, which can be negative.

As a result, the smallest nonzero singular value of the Hessian $\lambda_{\min}^{\neq 0}(\nabla_{yy}^2g(x,y))$ is always positive and $\lambda_g$ serves as the lower bound for a set of positive numbers, regardless of whether the eigenvalues are positive or nonpositive. Therefore, even if there are nonpositive eigenvalues, $\lambda_g>0$ can still hold. We will rephrase the sentences in the paper to reflect this.

**G3: The choices of $\alpha,\beta,\rho,c,N,\eta_1$.**

Due to the space limitation, we defer the choices of parameters in the Line 701 of Appendix. Let us define the condition number as $\kappa_1=\ell_{g,1}/\mu_{g}$ and $\kappa_2=\ell_{g,1}/\lambda_{g}$, we have
\begin{align*}
&\beta\leq\min\left\\{\frac{1}{\ell_{g,1}},\frac{c\mu_g^2}{16L_{F}\ell_{g,1}^2},\frac{c\ell_{g,1}^2}{\mu_g^2(L_{F}+2\ell_{g,1}L_w+c(L_g+\ell_{g,1}))}\right\\}={\cal O}(\min\\{\kappa_1,\kappa_2\\}), \\\\
&\alpha\leq \frac{\mu_g^2}{8c\ell_{g,1}^2}\beta={\cal O}\left(\frac{\min\\{\kappa_1,\kappa_2\\}}{\kappa_1^4\kappa_2^2}\right),\quad \rho\leq\frac{1}{\ell_{g,1}^2}, \quad c={\cal O}(\kappa_1^2\kappa_2^2), \quad\eta_1=2c\ell_{g,1}
\end{align*}
where $L_w=\frac{\ell_{f,1}}{\lambda_g}+\frac{\sqrt{2}\ell_{g,2}\ell_{f,0}}{\lambda_g^2},L_{F}=\frac{\ell_{f,0}(\ell_{f,1}+\ell_{g,2})}{\lambda_g}$ and $N\geq 1$ is any constant.

**G4：Can $w_0$ be initialized from other points?**

Thank you for this interesting questions which may help us extend our analysis.

**$w^{0}$ can be other points other than $0$ as long as $w^{k,0}=w^0$ for any $k$.** Even though the limit point $\lim_{t\rightarrow\infty}w^{k,t}$ is not the minimum-norm solution $w^\dagger(x^k,y^{k+1})$, it is bounded and Lipschitz continuous because
\begin{align}\lim\_{t\rightarrow\infty}w^{k,t}&=w^\dagger(x^k,y^{k+1})+\operatorname{Proj}\_{\operatorname{Ker}(\nabla_{yy}^2 g(x^k,y^{k+1}))}(w^0)\\\\&=w^\dagger(x^k,y^{k+1})+(I-(\nabla_{yy}^2 g(x^k,y^{k+1})^\dagger \nabla_{yy}^2 g(x^k,y^{k+1}))w^0.
\end{align}
Clearly, the limit point $\lim_{t\rightarrow\infty}w^{k,t}$ is also bounded. Moreover, it is Lipschitz continuous since for any $x_1,x_2$ and $y_1,y_2$, we have
\begin{align*}
&\quad\\|w^\dagger(x_1,y_1)+\operatorname{Proj}\_{\operatorname{Ker}(\nabla_{yy}g(x_1,y_1))}(w^0)-w^\dagger(x_2,y_2)-\operatorname{Proj}\_{\operatorname{Ker}(\nabla_{yy}g(x_2,y_2))}(w^0)\\|\\\\&\leq \\|w^\dagger(x_1,y_1)-w^\dagger(x_2,y_2)\\|+\\|\nabla_{yy}^2 g(x_1,y_1)^\dagger\\|\\|\nabla_{yy}^2 g(x_1,y_1)-\nabla_{yy}^2 g(x_2,y_2)\\|\\|w^0\\|+\\|\nabla_{yy}^2 g(x_1,y_1)^\dagger-\nabla_{yy}^2 g(x_2,y_2)^\dagger\\|\\|\nabla_{yy}^2 g(x_2,y_2)\\|\\|w^0\\|\\\\&\leq \left(L_w+\frac{\ell_{g,2}\\|w^0\\|}{\lambda_g}+\frac{\sqrt{2}\ell_{g,2}\ell_{g,1}\\|w^0\\|}{\lambda_g^2}\right)\\|(x_1,y_1)-(x_2,y_2)\\|.
\end{align*}
With the boundedness and Lipschitz continuity of limit points, we can prove the same convergence argument.

---

> ### Comment · Area_Chair_UtpA · 2023-08-10
> **Small clarification regarding eigenvalues**
>
> Dear authors, thanks for your careful rebuttal.
>
> This is a very minor point, but I was also baffled by Assumption 2, and your response here clarifies everything. Still, I think that using $\lambda$ as a notation for singular values is a bad idea; why not use the canonical notation $\sigma$? Or, even better, since everything here is symmetric, why not simply say that the non-zero eigenvalues are bounded away from 0? Either solution would make this assumption much clearer.
>
> Thanks,
> AC

---

> > ### Author Response · Authors · 2023-08-10
> > **Thank you for prompt reply**
> >
> > Thank you for your insightful advice and prompt reply！We agree that using $\sigma$ as the notation for the singular value is more suitable than $\lambda$, and we will definitely revise it in our revision.
> >
> > We look forward to the rolling dicussion and further engagement with the reviewers and area chair!
> >
> > Best,
> >
> > Your authors

---

> ### Comment · Reviewer_5R75 · 2023-08-14
> **Stationarity measure needs more justification**
>
> I realized that when you add the Hessian multiplication to the stationarity measure, you may not be solving the same problem anymore.
>
> The KKT condition of the original problem must satisfy
> $\nabla_y f(x^*, y^*) + w^* \nabla_{yy}^2 g(x^*, y^*) = 0$,
> but multiplying additional Hessian ignores the component of $\nabla_y f(x^*, y^*)$ that belongs to the kernel of $\nabla_{yy}^2 g(x^*, y^*)$.
>
> I think this part must be clarified. As you stated, it is only a necessary condition, and I do not think it sufficiently justifies the use of criteria to solve the original Bilevel problem. (Even if I give you the correct $x^*$, you might find a wrong $y$ where the condition $x$ is violated and start to move away from $x^*$).

---

> > ### Comment · Area_Chair_UtpA · 2023-08-14
> > **Re: non-zero eigenvalue assumption**
> >
> > Dear authors,
> >
> > In assumption 2, you assume that the non-zero eigenvalues are bounded away from 0. However, under sufficient regularity assumption, they will move smoothly with $x, y$. Therefore, I cannot see how there can ever be a negative eigenvalue since it would stay negative for all $x, y$, making each stationary point a saddle point and not a local minimum.
> >
> > Can you give an example of a problem satisfying assumption 2 with a negative eigenvalue?
> >
> >
> > Best,
> > AC

---

> > > ### Author Response · Authors · 2023-08-16
> > > **Re: Re: non-zero eigenvalue assumption**
> > >
> > > Dear AC and Reviewers,
> > >
> > > Thank you for your insightful question!
> > >
> > > As the major question is related to Assumption 2, we want to further clarify it. For ease and accuracy of use, we decompose Assumption 2 into three parts:
> > >
> > > **(2a) $g(x,y)$ satisfies PL condition;**
> > >
> > > **(2b) Non-zero singular value of $\nabla_{yy} g(x,y)$ bounded away from $0$ everywhere;**
> > >
> > > **(2c) Non-zero singular value of $\nabla_{yy} g(x,y)$ bounded away from $0$ on a bounded set.**
> > >
> > > With the above assumptions, the contributions of our paper lie in two aspects:
> > > 1) proving the inherent CQ condition and proposing the generalized stationary measure for PL bilevel problem in Lemma 3 and Theorem 1, which only require Assumption **(2a)**;
> > > 2) generalizing the alternating implicit gradient descent algorithm to achieve the $\epsilon$ - stationary point in Theorem 2, which requires both **(2a)** and **(2b)** in the submission.
> > >
> > > Our first contribution, grounded exclusively on Assumption **(2a)**, paves the way for future researchers to safely employ KKT-type measures for PL bilevel problems, ensuring they are free from CQ-invalid concerns.
> > >
> > > Second, it appears to us that **(2b)** together with **(2a)** will indeed exclude all negative eigenvalues due to Hessian Lipschitz continuity. In this case, **our convergence rate of GALET is still correct**, but it will be applicable in the convex lower-level (LL) case. We will **revise Assumption 2** and **rephrase our second claim** in the submission to the **bilevel with 'convex' LL** setting. It is worthy to notice that this is still more general than strongly convex case because we allow zero singular value and thus **multiple LL solutions**. Moreover, GALET still achieves the best-known convergence rate compared with [R2,R3] in convex bilevel literature.
> > >
> > > Finally, **(2b)** in our paper is used to ensure the boundedness of pseudo-inverse of the Hessian, and can be relaxed from the whole space to along the trajectory. As long as the trajectory remains bounded, **(2b)** can be reduced to **(2c)**, which can **include negative eigenvalues**. This can be achieved by the following procedure. S1) Assume the boundedness of pseudo-inverse of the Hessian near the LL optimal solutions is standard, e.g. in [2, Proposition 16] which is implied by Morse-Bott function and is used to prove the asympototic behavior of the proposed algorithm; and [33, Proposition B.1] (i.e. [R1, Proposition 6.3]) which is implied by constant rank CQ condition and is used to show the asymptotic relations between two stationary metrics. Therefore, similarly by relaxing Assumption 2 to merely near optimal points, we are able to attain the asymptotic convergence of GALET. S2) We additionally assume coerciveness like in [2], and we are able to prove the boundedness of the sequence $\{x^k,y^k\}$ generated by GALET. In this way, **(2b)** can be relaxed to **(2c)** for a bounded region of the starting point. Beyond this region, one can safely have negative eigenvalue.
> > >
> > > [R1] Gong, Chengyue, Xingchao Liu, and Qiang Liu. "Automatic and harmless regularization with constrained and lexicographic optimization: A dynamic barrier approach." Advances in Neural Information Processing Systems, 2021.
> > >
> > > [R2] Sow, Daouda, et al. "A Primal-Dual Approach to Bilevel Optimization with Multiple Inner Minima." arXiv preprint arXiv:2203.01123, 2022.
> > >
> > > [R3] Chen, Lesi, Jing Xu, and Jingzhao Zhang. "On bilevel optimization without lower-level strong convexity." arXiv preprint arXiv:2301.00712, 2023.
> > >
> > > We apologize for the confusion. We will rephrase the assumptions in the revised version and hope these clarifications have resolved your concerns!
> > >
> > > Many thanks,
> > >
> > > Your authors

---

> > > > ### Comment · Area_Chair_UtpA · 2023-08-16
> > > >
> > > > Dear authors,
> > > >
> > > > Thanks for your reply and clarification.
> > > >
> > > > I am still concerned by assumptions 2.b - 2.c.
> > > >
> > > > Indeed, they prevent the number of negative eigenvalues to change over the domain. Since at the optimal points in y, we cannot have a negative eigenvalue, these assumptions make it impossible to have a negative eigenvalue along the optimization trajectory, hence they can simply be rephrased as « all eigenvalues are >= 0, and the nonzero eigenvalues are bounded away from 0 ».
> > > >
> > > > Is this correct ?
> > > >
> > > > Best,
> > > > AC

---

> > > > > ### Author Response · Authors · 2023-08-16
> > > > > **Re: Official Comment by Area Chair UtpA**
> > > > >
> > > > > Dear AC,
> > > > >
> > > > > Thanks for your prompt response!
> > > > >
> > > > > Yes, you are right that we can simply rephrase (2b) as  "*all eigenvalues are nonnegative, and the nonzero eigenvalues are bounded away from 0*." This setting will reduce to the bilevel with convex LL case admitting *multiple LL solutions*.
> > > > >
> > > > > In addition, we can also consider relaxing the assumption to include negative eigenvalues, such as (2c) (in a bounded domain) or along the discrete optimization trajectory, in which case it only prevents negative eigenvalue near the convergence point (the dense region with small stepsizes and small gradients); see the two-step procedure in our previous response.
> > > > >
> > > > > Many thanks,
> > > > >
> > > > > Your authors

---

> > ### Author Response · Authors · 2023-08-16
> > **Re: Stationarity measure needs more justification**
> >
> > Dear Reviewer 5R75,
> >
> > Thank you for your prompt reply!
> >
> > In fact, adding Hessian does not change the original bilevel problem. In our paper, the logic unfolds in two main steps. First, we identify a tight necessary condition for optimality in the original bilevel problem, which serves as the stationary measure (counterpart of $\nabla_x f(x,y^*(x))=0$ in the strongly-convex bilevel case). Following this, we propose an algorithm that converges to an $\epsilon$-stationary point, in accordance with our specific definition.
> >
> > We have proven that (8) with additional Hessian is a necessary condition for the global optimality in the original bilevel problem in Theorem 1. Moreover, it is tight in the sense that it not only recovers the stationary metric in a strongly convex case, but also recovers the stationary measure in the Morse-Bott case [2]; See Appendix H.1. Therefore, (8) is informative to the original bilevel problem.
> >
> > As for your question 'even if I give you the correct $x$, you might find a wrong $y$ where the condition (8a) is violated and start to move away from $x^*$', this is because the ultimate goal is to find the optimal $(x^*,y^*)$ pair, but we have not found the optimal $y$ yet. The same phenomenon also occurs in strongly-convex BLO as our stationary measure is a generalized one. For example, for fully single loop BLO algorithm [11,31], if $x=x^*$ but $y$ is not optimal and even if $w=-\nabla_{yy}g(x,y)^{-1}\nabla_y f(x,y)$ is exact, updating $x$ through $\nabla_x f(x,y)-\nabla_{xy}g(x,y)\nabla_{yy}g(x,y)^{-1}\nabla_y f(x,y)$ might also cause $x$ starting to move away from $x^*$.
> >
> > For the ease of your convenience, we copy the statement that (8) can be reduced to the stationary measure in strongly-convex and Morse-Bott function in Appendix H.1 here.
> >
> > **1. Nonconvex-strongly-convex BLO or PL with invertible Hessian.** In this case, $S(x)$ is a singleton for any $x$ and $\nabla_{yy}g(x,y)$ is always non-singular so that the solution to (8b) is uniquely given by
> > \begin{align}
> >     w^*=-\left(\nabla_{yy}^{2}g(x^*,S(x^*))\right)^{-1}\nabla_y f(x^*,S(x^*)).
> > \end{align}
> > Therefore, the necessary condition in (8) is equivalent to
> > \begin{align}
> >     \nabla_x f(x^*,S(x^*))-\nabla_{xy}^2g(x^*,S(x^*))\left(\nabla_{yy}^{2}g(x^*,S(x^*))\right)^{-1}\nabla_y f(x^*,S(x^*))=0
> > \end{align}
> >
> > Therefore, by the implicit function theorem, (8) recovers the necessary condition $\nabla f(x^*,S(x^*))=0$ for nonconvex-strongly-convex and PL with invertible Hessian BLO.
> >
> > **2. Nonconvex-Morse-Bott BLO.** Morse-Bott functions are a special case of PL functions [2]. In this case, $\nabla_{yy}g(x,y)$ can be singular so that (8b) may have infinitely many solutions. Letting $\operatorname{Ker}(A)$ denote the null space of the matrix $A$, the solution set of (8b) is given by
> > \begin{align*}
> >     {\cal W}^*=-\left(\nabla_{yy}^2 g(x^*,y^*)\right)^\dagger\nabla_y f(x^*,y^*)+\operatorname{Ker}(\nabla_{yy} ^2g(x^*,y^*)).
> > \end{align*}
> > where $\left(\nabla_{yy}^2 g(x^*,y^*)\right)^\dagger$ is the pseudoinverse of the Hessian. According to [2, Proposition 6], it holds that for any $y^*\in S(x^*)$, $\nabla_{xy}^2g(x^*,y^*)$ belongs to the range of $\nabla_{yy}^2g(x^*,y^*)$ so that $\nabla_{xy}^2g(x^*,y^*)$ is orthogonal to $\operatorname{Ker}(\nabla_{yy}^2 g(x^*,y^*))$. As a result, although the solution to (8b) is not unique, all of possible solutions yield the unique LHS value of (8a). i.e. $\forall w^*\in{\cal W}^*$,
> > \begin{align*}
> >     \nabla_{xy}^2g(x^*,y^*)w^*=-\nabla_{xy}^2g(x^*,y^*)\left(\nabla_{yy}^2 g(x^*,y^*)\right)^\dagger\nabla_y f(x^*,y^*).
> > \end{align*}
> >
> > Plugging into (8a), we arrive at
> >
> > \begin{align*}
> >     \nabla_x f(x^*,y^*) -\nabla_{xy}^2g(x^*,y^*)\left(\nabla_{yy}^2 g(x^*,y^*)\right)^\dagger \nabla_y f(x^*,y^*)=0 \text{ and }\nabla_y g(x^*,y^*)=0
> > \end{align*}
> > where $\phi(x^*,y^*):=-\nabla_{xy}^2g(x^*,y^*)\left(\nabla_{yy}^2 g(x^*,y^*)\right)^\dagger$ is the same as the degenerated implicit differentiation in [2]. This replaces $\left(\nabla_{yy}^{2}g(x^*,y^*)\right)^{-1}$ in strongly-convex case by the pseudoinverse $\left(\nabla_{yy}^2g(x^*,y^*)\right)^{\dagger}$.
> >
> >
> > We hope the detailed clarifications can fully resolve your concerns!
> >
> > Many thanks,
> >
> > Your authors

---

### Decision · Program_Chairs · 2023-09-21

**Decision:**

Accept (poster)

**Comment:**

This paper tackles the hard problem of bilevel optimization with a non-strongly convex inner problem.

The proposed algorithm is simple and natural, and its convergence is proven, obtaining the same rate as gradient descent in single-level optimization.

The main problem with this paper raised by rev.k2BT is the assumption that all eigenvalues of the hessian are either 0 or bounded away from 0. The authors claim in the original version that this is stronger than simply assuming that the eigenvalues are either 0 or positive and bounded away from 0, which is a classical assumption in several other papers.

However, by continuity, the two assumptions are the same; there cannot be a negative eigenvalue if the eigenvalues cannot change sign. The authors acknowledge this in the discussion. This assumption should, therefore, be rewritten in the final version, as well as the statement in l.277 "so that it 277 is weaker than [24]" which becomes false (although [24] has exactly the same bug in the assumption, the negative eigenvalues cannot suddenly become positive...). I also recommend that the authors clarify the notation regarding eigen / singular values, as this was a major source of confusion: singular values should not be written as $\lambda$.

This minor technical issue does not overshadow the merits of this paper: the problem tackled is hard, the contributions are original, the analysis is sound, and the experiments are conclusive.

Only Reviewer k2BT still advocates for the rejection of the paper; all other reviewers agree that the paper is worthy of acceptance. The concerns of Reviewer k2BT mostly revolve around the previous minor technical point. The authors have responded to these concerns in a convincing way, and the reviewer did not argue further with the authors and focused on this minor point in the reviewer's discussion.